# A megatransposon drives the adaptation of *Thermoanaerobacter kivui* to carbon monoxide

Rémi Hocq [1,2,3], Josef Horvath [1,2], Maja Stumptner[1,2], Mykolas Malevičius [4], Gerhard G. Thallinger [4] & Stefan Pflügl [1,2] ✉

Acetogens are promising industrial biocatalysts for upgrading syngas, a gas mixture containing CO, $H_2$ and $CO_2$ into fuels and chemicals. However, CO severely inhibits growth of many acetogens, often requiring extensive adaptation to enable efficient CO conversion (carboxydotrophy). Here, we adapt the thermophilic acetogen *Thermoanaerobacter kivui* to use CO as sole carbon and energy source. Isolate CO-1 exhibits rapid growth on CO and syngas (co-utilizing CO, $H_2$ and $CO_2$) in batch and continuous cultures ($\mu_{max}$ ~ 0.25 h$^{-1}$). The carboxydotrophic phenotype is attributed to the mobilization of a CO-dependent megatransposon originating from the locus responsible for auto-trophy in *T. kivui*. Transcriptomics reveal the crucial role the redox balance plays during carboxydotrophic growth. These insights are exploited to rationally engineer *T. kivui* to grow on CO. Collectively, our work elucidates a primary mechanism responsible for the acquisition of carboxydotrophy in acetogens and showcases how transposons can orchestrate evolution.

Synthesis gas (syngas), a mixture composed of hydrogen ($H_2$), carbon dioxide ($CO_2$), and carbon monoxide (CO) in variable proportions[1], serves as a fuel and chemical precursor for either catalytic or gas fermentation processes. Plant biomass as a renewably available resource is deemed essential for establishing a global circular carbon economy[2] and is a promising feedstock to sustainably generate syngas. Industrial syngas fermentation can be efficiently carried out by acetogens[3], a phylogenetically diverse group of bacteria. These microbes naturally ferment $H_2$ and $CO_2$ through the reductive acetyl-CoA pathway (or Wood-Ljungdahl pathway, WLP)[4,5], an ancient pathway that is thought to be intricately linked with the origin of life[6]. Within the WLP, CO plays a critical role as a key intermediate for acetyl-CoA formation. Being a possibly abundant energy source in primordial environments, CO may have been instrumental as an inorganic substrate for early life[7].

As a prerequisite for syngas fermentation, acetogens often need to undergo adaptation or evolution to tolerate and efficiently utilize CO (a lifestyle also known as carboxydotrophy), in addition to $H_2$ and $CO_2$[8–10].

In acetogens, CO toxicity is thought to be primarily caused by inhibition of hydrogenases crucial for redox metabolism[11], reducing or even abolishing growth. Among these hydrogenases, the [FeFe] hydrogen-dependent $CO_2$ reductase (HDCR) from *Acetobacterium woodii* and, to a much lower extent, the [NiFe] energy-converting hydrogenase (Ech) from *Thermoanaerobacter kivui* were shown to be inhibited by CO[12,13]. Circumventing this inhibition was demonstrated to be essential to promote growth of *A. woodii* on CO[14]. On the other hand, the $CO_2$/CO reduction potential is particularly low (E°′ ~ − 520 mV[15]), and the resulting low potential electrons stemming from CO oxidation can boost acetogenic metabolism by significantly increasing ATP yields compared to classical growth on $H_2$/$CO_2$[14,16]. In turn, higher ATP yields critically provide the opportunity to generate more ATP-intensive products such as short or medium-chain alcohols from syngas, or directly from CO (from industrial off-gas streams or obtained via $CO_2$ electrolysis)[17].

In the thermophilic acetogen *T. kivui*, CO metabolism has been investigated following adaptation[8,18]. CO enters the carbonyl branch of

[1]Institute of Chemical, Environmental and Bioscience Engineering, Technische Universität Wien, Gumpendorfer Straße 1a, 1060 Vienna, Austria. [2]Christian Doppler Laboratory for Optimized Expression of Carbohydrate-active Enzymes, Institute of Chemical, Environmental and Bioscience Engineering, TU Wien, Gumpendorfer Straße 1a, 1060 Vienna, Austria. [3]Circe Biotechnologie GmbH, Vienna, Austria. [4]Institute of Biomedical Informatics, Graz University of Technology, Graz, Austria. ✉e-mail: stefan.pfluegl@tuwien.ac.at

the WLP and is bound to the acetyl-CoA synthetase. The methyl group required for acetyl-CoA synthesis is generated by the partial oxidation of CO to $CO_2$ via a monofunctional CO dehydrogenase (CODH) and subsequent $CO_2$ reduction in the methyl branch of the WLP (which notably involves the HDCR), which yields methyl-CoFeSP. During carboxydotrophic growth, CO also serves as an energy source. Reduced ferredoxin ($Fd^{2-}$) generated during CO oxidation by the CODH simultaneously drives the formation of $H_2$ as well as of a proton gradient via the Ech complex, which is subsequently used for ATP synthesis. Although the metabolic model and the bioenergetics of *T. kivui* have not yet been fully elucidated, current metabolic models indicate the ATP yield is much higher on CO than on $H_2/CO_2$ (~ 1.37 vs 0.27 $mol_{ATP}\ mol_{acetate}^{-1}$, assuming the methylene-THF dehydrogenase/Ech complex couples the generation of methyl-THF to the export of two protons from the cell[19,20]).

Previous work suggested that the buildup of CO tolerance and utilization in *T. kivui* might be linked to mutations occurring at the HDCR, potentially improving its tolerance towards CO. However, these mutations were not functionally characterized[21], and whether they indeed promote carboxydotrophy in *T. kivui* still remains to be elucidated. Carboxydotrophic *T. kivui* strains previously obtained were, in addition, shown to display low growth rates on CO as the sole carbon and energy source ($T_d$ ~ 40 h/$\mu_{max}$ = 0.017 $h^{-1}$)[8], severely limiting the potential of *T. kivui* for syngas bioprocessing. This low growth on CO in a chemically defined medium (i.e., without yeast extract) likely stems from the fact that adaptation to CO was performed in a medium supplemented with yeast extract, which is actively used as a substrate for growth by *T. kivui*[22]. As a result, evolved mutants not only strengthened their ability to grow on CO but also on yeast extract[22], and the omission of yeast extract might, therefore, severely limit growth rates. Ultimately, this unintended selection bias also complicates the functional analysis of the mechanisms underlying adaptation to carboxydotrophy, as mutations may potentially be linked to increased utilization of both carbon/energy sources.

Here, we create a robust chassis for thermophilic bioprocessing of syngas[23] and elucidate the cognate mechanisms allowing for CO tolerance and utilization. We combine adaptive laboratory evolution (ALE)[24], bioprocess engineering, and multi-omics data analysis to generate a strain rapidly growing on CO as the sole carbon and energy

source ($T_d$ ~ 2.8 h/$\mu_{max}$ = 0.25 $h^{-1}$), as well as to quantitatively and qualitatively evaluate its physiological, genetic and metabolic features. We show that in this strain, the acquisition of carboxydotrophy is intrinsically linked to the mobilization of a circular, CO-dependent, 86 kb megatransposon originating from the WLP locus. We further speculate that the synergistic up- and down-regulation of central redox metabolic genes is the primary effector of the acquisition of carboxydotrophy, by alleviating CO-mediated ferredoxin overreduction. Finally, we validate this theory by enabling carboxydotrophic growth in the WT by overexpressing the energy-converting hydrogenase Ech2 complex.

## Results

### Wild-type *T. kivui* can efficiently grow on CO as sole carbon and energy source

*Thermoanaerobacter kivui* DSM 2030 was adapted to carboxydotrophic growth by an ALE approach featuring three phases (Fig. 1a). First, the DSM 2030 strain was cultivated on $H_2$ and $CO_2$ to establish robust autotrophic growth in chemically defined mineral medium (without yeast extract and vitamins), yielding a cell population referred to as $WT_{pop}$. Next, two syngas mixtures containing different concentrations of $H_2$, $CO_2$, CO and $N_2$ (either 58:9:30:3 or 24:21:52:3, 2 bar) were separately used to adapt *T. kivui* to the presence of CO. Growth on CO as sole carbon and energy source (30% CO, 70% $N_2$, 2 bar) was achieved by using cells adapted to the 52% CO syngas mixture ($52S_{pop}$). Cells adapted to 30% CO syngas ($30S_{pop}$) were unable to grow on 30% CO. The CO concentration was gradually increased in 10% increments to 60% CO, and finally to 100% CO ($CO_{pop}$). From the first inoculation on 52% CO syngas, the $CO_{pop}$ culture (robustly growing on 2 bars 100% CO) was obtained in 31 generations.

Clonal strains from $WT_{pop}$, $52S_{pop}$, and $CO_{pop}$ cultures were isolated on plates, and the growth of cell populations and clonal strains on various gas mixtures was assessed (Fig. 1b). Growth on $H_2/CO_2$ and low CO syngas was possible for all populations and clones. Growth on high CO syngas was observed for all cultures except for the $WT_{pop}$ and the derived G-1 isolate. $52S_{pop}$ and related H-1 isolate, as well as $CO_{pop}$ and derived CO-1 clone, could all grow on 100% CO. Strain H-2 derived from the $52S_{pop}$ population, could interestingly not grow on CO.

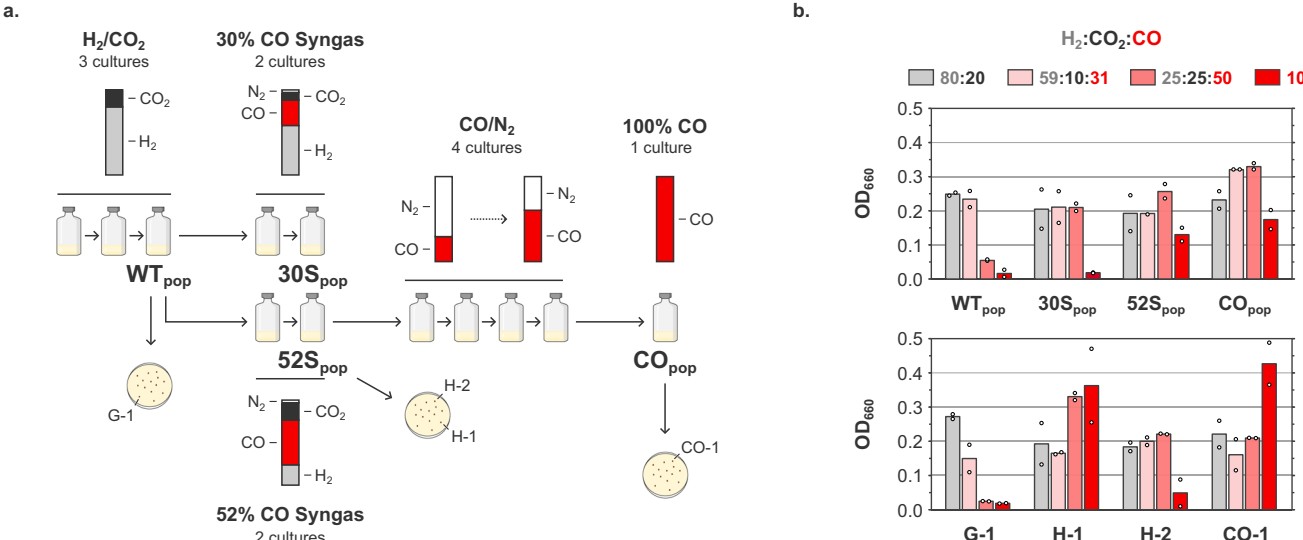

**Fig. 1 | CO adaptation strategy. a** *T. kivui* DSM 2030 was adapted to CO using serial serum bottle cultures with chemically defined mineral medium (without yeast extract and vitamins) under increasing CO concentrations. Clonal strains were isolated on plates from different cultures. **b** Biomass formation of adapted populations and clones after 3 days in serum bottles (chemically defined mineral medium without yeast extract and vitamins) under various gas mixtures, starting from $OD_{660}$ = 0.01. Source data are provided as a Source Data file.

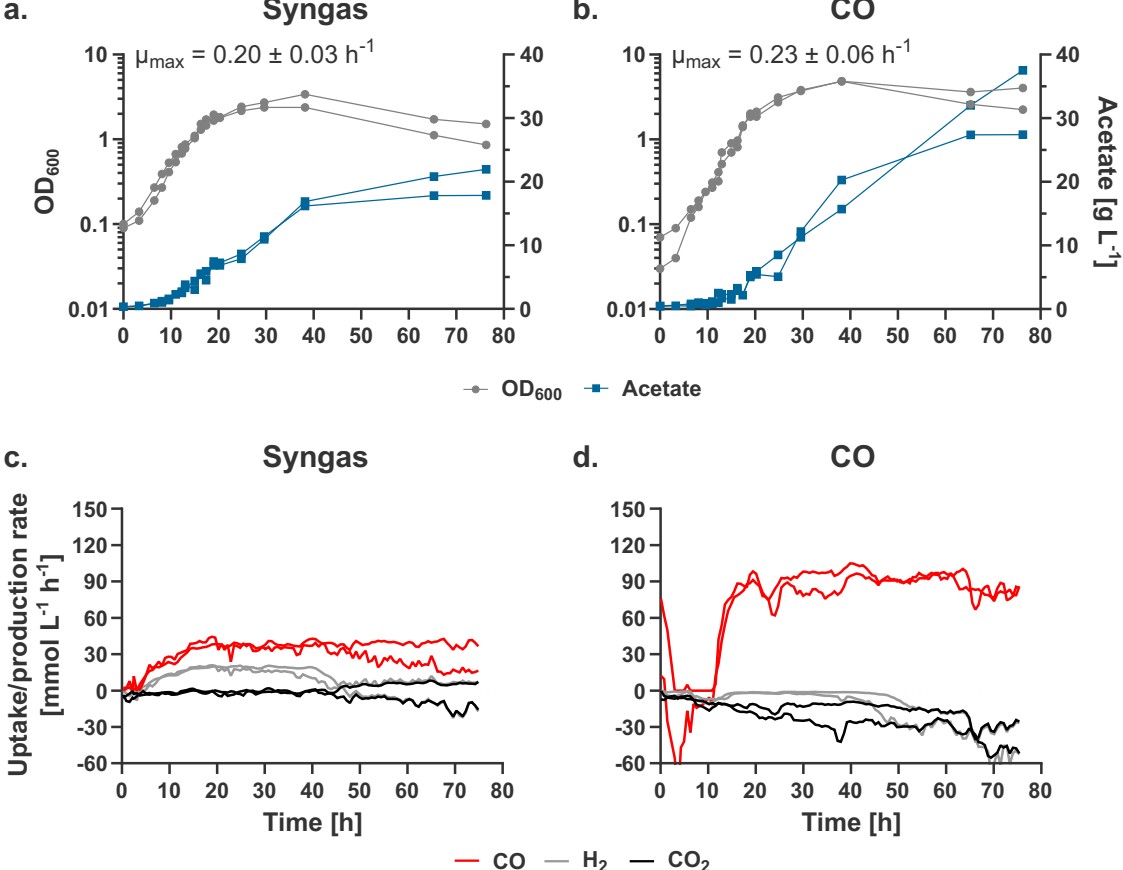

**Fig. 2 | Batch gas fermentation of CO-1 in bioreactor.** A synthetic syngas mixture ($CO/CO_2/H_2/N_2$: 52/21/24/3) and 100% CO were used as the carbon and energy source in chemically defined mineral medium (without yeast extract and vitamins). **a**, **b** Biomass ($OD_{600}$), acetate formation, and maximum specific growth rate on syngas (**a**) and CO (**b**). **c**, **d.** Gas uptake (positive) and production (negative) rates on syngas (**c**) and CO (**d**). Change of gas uptake/production rates at the end of the experiment due to flushing with $N_2$. The maximum specific growth rate $\mu_{max}$ ($\pm$ standard deviation) was calculated from the exponential growth phase of each culture (6–13 h for syngas and 13–19 h for CO). Source data are provided as a Source Data file.

Collectively, our results indicate that the pure carboxydotrophic trait, allowing for growth on CO as sole carbon and energy source, was present already at the $52S_{pop}$ stage, and therefore likely arose between that culture and the $WT_{pop}$ culture. Only two subculturing steps chronologically separate these cultures (~ 8-9 generations), suggesting that such a rapid acquisition of the carboxydotrophic trait more likely stemmed from adaptation rather than evolution.

### CO-1 shows high fermentative performance on syngas and CO in bioreactor batch cultivation

To assess the potential of CO-1 for syngas conversion and consolidate its physiological characterization, we evaluated CO-1 fermentative capabilities on two different gas mixtures ($CO/H_2/CO_2/N_2$: 52/24/21/3 and 100% CO) in mineral medium in bioreactor batch cultivations with continuous gassing (Fig. 2). The maximum growth rate $\mu_{max}$ reached $0.20 \pm 0.03\,h^{-1}$ ($T_d$ ~ 3.5 h) and $0.23 \pm 0.06\,h^{-1}$ ($T_d$ ~ 3.0 h) on syngas and CO, respectively. Wild-type *T. kivui* does not grow under either of these conditions. High biomass was formed in both conditions ($OD_{600}$ of $2.89 \pm 0.72$ and $4.85 \pm 0.01$ for syngas and 100% CO conditions, respectively), and acetate titers reached high levels ($19.9 \pm 2.9$ and $32.5 \pm 7.1\,g\,L^{-1}$). Importantly, CO and $H_2$ were co-consumed in the syngas condition, which is a critical aspect for syngas bioprocessing. Under 100% CO, later stages of fermentation show a water-gas shift reaction, resulting in the net production of $H_2$ in addition to $CO_2$. Together, the results of our batch fermentations confirm that CO-1 is particularly well-suited for syngas and CO conversion.

### Short-read sequencing analysis reveals only a few SNVs/InDels in the CO-adapted strains

Although *T. kivui* was suspected to have gained the carboxydotrophic trait through adaptation rather than evolution, we submitted the CO-adapted clonal strains for whole genome sequencing (Illumina). These encompass CO-1, as well as two other carboxydotrophic isolates from $CO_{pop}$ (CO-2, CO-3) with a similar phenotype, albeit growing at a lower growth rate (Supplementary Fig. 1). DNA from the non-carboxydotrophic $WT_{pop}$ was sequenced as a reference. As expected, only a few SNVs and InDels were found in the CO-adapted strains (total number across the three strains: 22, Supplementary Data 1). Common mutations in all three strains (Table 1) do not directly affect genes involved in CO metabolism, rendering their potential contribution to the carboxydotrophic phenotype unclear.

### Presence of a large extrachromosomal transposable element in the CO-1 strain

Short-read sequencing can also be used to gain insights into potential large-scale rearrangements, and we therefore performed coverage analysis in the CO-adapted strains. A 2438 bp deletion was found at positions 233,431 to 235,869, affecting two genes with unknown function as well as an additional one annotated as N-acetyltransferase (TKV_RS01120 to TKV_RS01130). In addition, two large duplications were found at positions 1,839,957 to 1,900,058 (duplication 1) and 1,908,405 to 1,934,113 (duplication 2), spanning 60,101 and 25,708 bp respectively (Fig. 3a). These duplications both display ~ 2.2 times the coverage of the rest of the genome (1765x and 1739x, respectively,

**Table 1 | Common mutations found in CO-1, CO-2 and CO-3 strains**

| Genome position | Type | Locus tag | Effect | Annotation |
|---|---|---|---|---|
| 625,773 | Substitution | TKV_RS07415 | D215N (GAT → AAT) | Phosphate signaling complex protein PhoU |
| 1,027,204 | Substitution | Intergenic | Intergenic (T → A) | |
| 1,266,481 | Substitution | TKV_RS04105 | A93P (GCT → CCT) | 4-hydroxy-tetrahydrodipicolinate synthase DapA |
| 1,963,781 | Substitution | TKV_RS10045 | D187E (GAT → GAA) | Elongation factor Tu |
| 1,968,524 | Substitution | TKV_RS10070 | A944D (GCT → GAT) | DNA-directed RNA polymerase subunit beta' |

Positions refer to wild-type *T. kivui* (GCA_963971585.1).

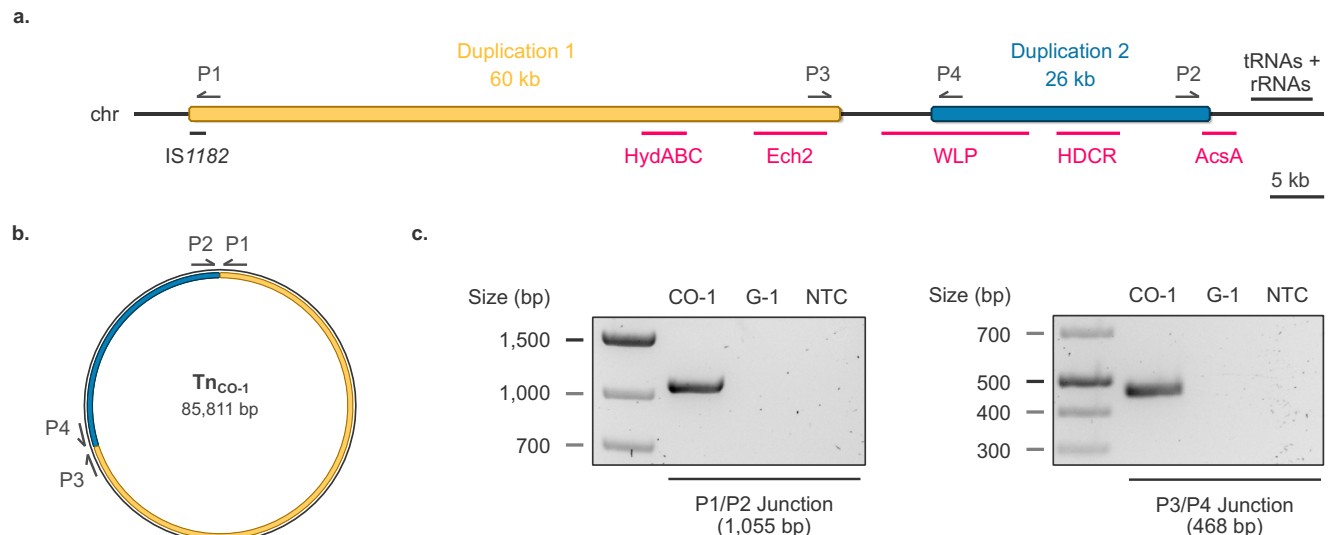

**Fig. 3 | CO−1 large scale genome rearrangement identified by WGS. a** Coverage analysis (Illumina) highlights two regions (positions 1,839,957–1,900,058, duplication 1 and 1,908,405–1,934,113, duplication 2) duplicated in CO-1 encompassing most of the WLP locus as well as other gene clusters important for acetogenesis. **b** Long read sequencing (ONT) reveals that both duplicated regions form a large, single, circular extrachromosomal element (Tn$_{CO-1}$) containing an IS$1182$-like transposase gene. **c** Validation of the presence of Tn$_{CO-1}$ in CO-1 DNA by PCR. Tn$_{CO-1}$-specific junctions were amplified using P1/P2 and P3/P4 primers. NTC: no template control. The validation of the presence of Tn$_{CO-1}$ was performed as a single experiment. Source data are provided as a Source Data file.

compared to 802x). The duplications are separated by 8348 bp and bear genes crucial for acetogenesis, namely, genes coding for the HDCR, Ech2, and HydABC complexes, as well as a significant part of the WLP locus (Fig. 3a). Duplication 1 starts with a gene encoding for an IS$1182$-type transposase, while duplication 2 ends just before a region rich in tRNA and rRNA genes, which are typical recognition sites for bacterial transposases[25–33]. This disposition, as well as the increased coverage strongly suggested that the large genomic region encompassing the WLP locus could be mobilizable by transposition elements. We, therefore, submitted DNA extracted from CO-1 to long-read sequencing (Oxford Nanopore Sequencing, ONT) to pinpoint the localization of the additional copies of duplications 1 and 2. De novo assembly yielded a circular 2,396,992 bp sequence identified as *T. kivui* chromosome (WT chromosome: 2,397,805 bp), as well as an additional circular sequence of 85,811 bp. The chromosome still bears an intact WLP locus, while the extrachromosomal element, designated as Tn$_{CO-1}$, links both duplications as depicted in Fig. 3b. Junction sequences linking both duplications are specific to Tn$_{CO-1}$ and primer pairs amplifying these short Tn$_{CO-1}$-specific sequences (which cannot be found on the genome) could therefore be designed (Fig. 3a, b). Successful PCR amplification was observed for CO-1 gDNA, but not G-1 gDNA, confirming the presence of Tn$_{CO-1}$ in vivo (Fig. 3c). Additional transposon activity was found when aligning CO-1 and WT chromosomes, with an insertion of an ISL3 family transposase and a translocation of an IS$Lre2$-family transposase, neither affecting genes involved in carboxydotrophy (Supplementary Table 1).

Tn$_{CO-1}$ presence was next assessed in CO-2, CO-3, H-1, and H-2 strains by searching specifically for junction sequences, defined as 50 bp sequences linking the last 25 bp of each duplicated sequence. Tn$_{CO-1}$ could be found in all strains able to grow on CO (CO-2, CO-3, H-1) but was absent from strains unable to grow on CO (H-2). As a control, we also checked the presence of Tn$_{CO-1}$ in the WT$_{pop}$ population. Expectedly, we did not find corresponding reads. On the other hand, de novo assembly of the H-2 genome showed that it bears the 2438 bp deletion found in CO-1 (similarly to CO-2, CO-3, and H-1), indicating that this deletion probably does not play an important role in adaptation to CO.

**Tn$_{CO-1}$ mobilization is dependent on CO**
When changing the carbon source to glucose, we noticed long lag phases that were also observed when reverting from glucose to CO. We hypothesized this could be due to a dynamic regulation of Tn$_{CO-1}$ favoring or impeding growth depending on the carbon/energy source. To test this hypothesis, we first performed semi-quantitative PCR targeting transposon junctions on DNA extracted from CO-1 and G-1 after a single preculturing step on glucose, H$_2$/CO$_2$ and CO (the latter for CO-1 only, Fig. 4a). Abundance of Tn$_{CO-1}$-specific amplicons markedly increased when cells were grown on CO, indicating a higher prevalence of the megatransposon in this condition. The abundance was also higher on glucose than on H$_2$/CO$_2$. As only one preculturing/culturing step separates the glucose and H$_2$/CO$_2$ cultures, the lower abundance on H$_2$/CO$_2$ might be linked with a faster decline in Tn$_{CO-1}$ copies than

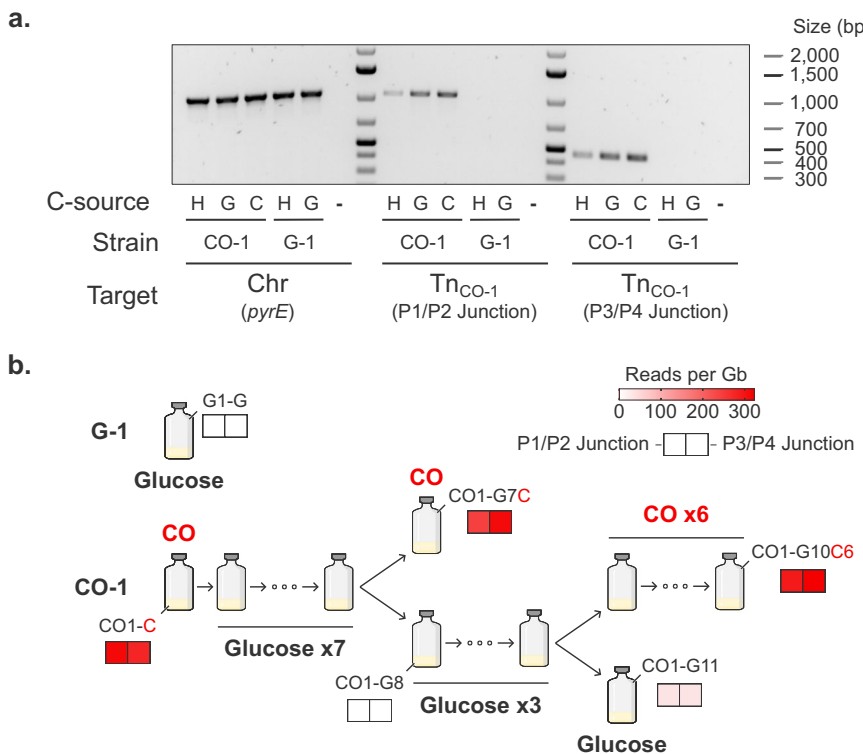

**Fig. 4 | Tn$_{CO-1}$ mobilization is CO-dependent. a** Semi-quantitative end-point PCR was performed on gDNA extracted from CO-1 and G-1 strains grown under different carbon sources. H: 2 bar H$_2$/CO$_2$ (80:20). G: 5 g L$^{-1}$ glucose. C: 2 bar CO. Shown are a PCR targeting chromosomal DNA (*pyrE* gene), as well as both Tn$_{CO-1}$-specific junctions. -: no template control. The end-point PCR was performed as a single experiment. **b** Adaptation to glucose and readaptation to CO was monitored using deep sequencing data (ONT). Tn$_{CO-1}$ presence was assessed by quantifying the abundance of Tn$_{CO-1}$-specific junction sequences. Source data are provided as a Source Data file.

on glucose. This could indeed be the case if the mobilization of Tn$_{CO-1}$ had a higher fitness cost on H$_2$/CO$_2$ – an energetically limited substrate - than on glucose.

Tn$_{CO-1}$ is extrachromosomal, and a decrease in abundance over time on alternative carbon sources could potentially lead to a complete loss of carboxydotrophy. We, therefore, subcultured CO-1 repeatedly on glucose and assessed whether the strain was able to grow again on CO after seven or ten culturing steps (Fig. 4b). Readaptation to CO was successful, and cells at various stages of the lineage were sequenced using ONT. Tn$_{CO-1}$ abundance was quantified (normalizing by sequencing depth) and shown to be, at best very low when cells were grown on glucose (between 0 and 34 reads per Gb). In contrast, re-exposure to CO resulted in much higher levels of Tn$_{CO-1}$-specific reads (~300 reads per Gb), even after prolonged culturing on glucose. In this experiment, mobilization of Tn$_{CO-1}$ was therefore either strongly CO-inducible and/or selected upon exposure to CO, which provides additional evidence of its importance for carboxydotrophy.

**Carboxydotrophic growth is abolished in the absence of Tn$_{CO-1}$**
Wild-type *T. kivui* is naturally competent[34], which makes the strain genetically tractable. As we recently developed a CRISPR-based genome editing system for *T. kivui*[35], we sought to gain additional insights into the putative role Tn$_{CO-1}$ plays in the carboxydotrophic phenotype of CO-1 by targeting the Tn$_{CO-1}$-specific junctions. However, our initial attempts to transform the CO-1 strain were unsuccessful, and we adapted the transformation protocol by selecting neomycin-resistant cells in liquid before plating to circumvent low plating efficiency. Using this approach, we managed to isolate a CO-1 transformant bearing an empty backbone (Fig. 5a). Surprisingly, the resulting strain was no longer able to grow on CO and lost most of its

capacity to grow on high CO syngas (H$_2$/CO$_2$/CO: 25/25/50, Fig. 5b). DNA was extracted from cells grown on low CO syngas (H$_2$/CO$_2$/CO: 60/15/25, a mixture also allowing low growth of the WT) and genotyping further showed that transformation of CO-1 resulted in the loss of Tn$_{CO-1}$ (Fig. 5c). This was confirmed by long-read sequencing, which failed to detect any reads indicative of the presence of Tn$_{CO-1}$, while confirming all CO-1-specific mutations to be present. This observation eliminated the need to use the CRISPR system to delete Tn$_{CO-1}$, as transformation alone seems to be incompatible with its mobilization. Although the underlying reason remains unclear, one possibility is that Tn$_{CO-1}$ negatively affects competency. In this scenario, transformation may have selected a clone lacking Tn$_{CO-1}$. Taken together, our results, therefore, unambiguously show that the highly efficient carboxydotrophic lifestyle displayed by CO-1 is solely linked to the mobilization of the megatransposon Tn$_{CO-1}$.

**The CO-1 strain can efficiently convert CO-containing gasses in chemostat cultures**
To characterize the physiology and study gene expression levels of strain CO-1, we established chemostat gas fermentations. This system provides well-defined, quantitative, stable, steady-state conditions, and enables precise control of the specific growth rate in bioreactors. Triplicate continuous cultures of G-1 and CO-1 were set up in 0.25 L bioreactor vessels with continuous gassing, adjusting stirring speed to reach about 2 g L$^{-1}$ acetate in the fermentation broth for H$_2$/CO$_2$ and CO and ~6 g L$^{-1}$ for syngas. G-1 and CO-1 were both cultivated on H$_2$/CO$_2$ and on syngas (H$_2$/CO$_2$/CO/N$_2$: 58/9/30/3), and CO-1 additionally on CO as the sole carbon and energy source at two different specific growth rates.

Specific gas consumption and acetate production rates, as well as biomass and acetate yields, were quantified for all conditions (Table 2). On H$_2$/CO$_2$ with a dilution rate D of 0.10 h$^{-1}$, acetate production yields

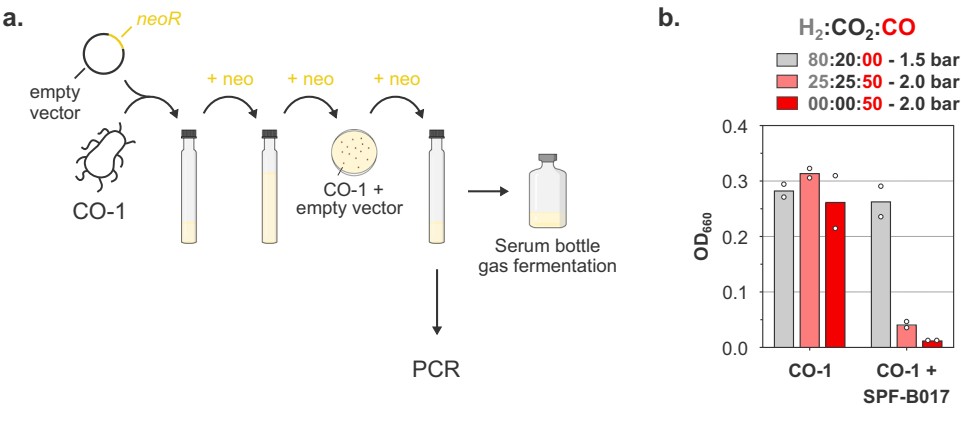

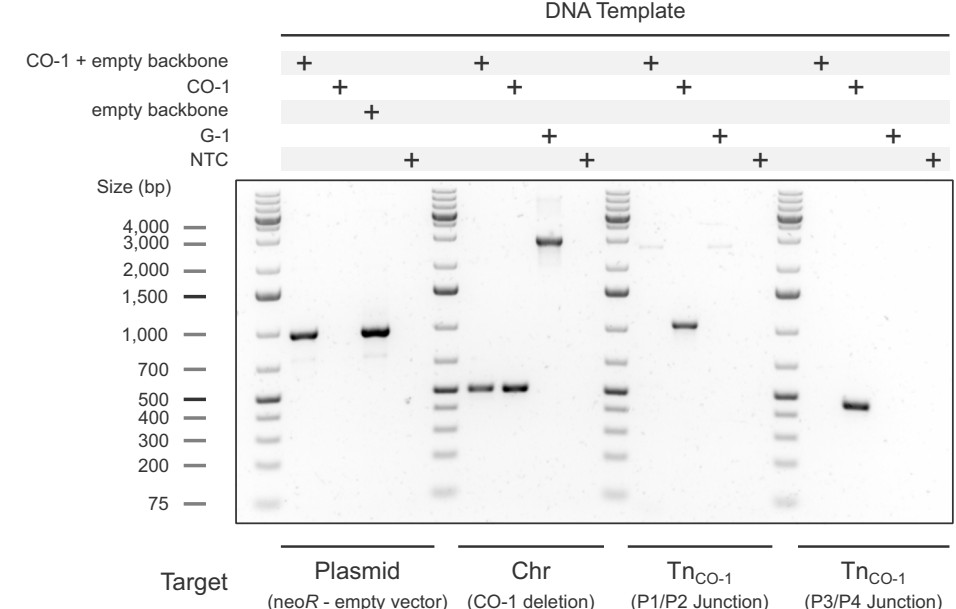

**Fig. 5 | Transformation of CO-1 results in loss of carboxydotrophy. a** Scheme depicting a transformation procedure adapted to the low transformation efficiency of CO-1. CO-1 was transformed with the empty SPF-B017 shuttle plasmid. **b** Biomass formation of non-transformed and transformed CO-1 after 3 days at 55 °C in serum bottles (mineral medium) under various gas mixtures. **c** PCR assay on DNA extracted from non-transformed and transformed CO-1. Shown are PCR targeting SPF-B017, a CO-1-specific deletion, as well as both $Tn_{CO-1}$-specific junctions. NTC: no template control. The genotyping of non-transformed and transformed CO-1 was performed as a single experiment. Source data are provided as a Source Data file.

were similar between CO-1 and G-1, with about 4 $H_2$ and 2 $CO_2$ consumed per acetate formed, as expected from typical acetogenesis[36]. On syngas, G-1 and CO-1 co-consumed $H_2$, CO, and $CO_2$ with similar specific uptake rates. On CO ($D = 0.10\,h^{-1}$), high specific CO consumption rates were associated with high biomass yields, which might be linked to higher ATP yields from the carboxydotrophic lifestyle[11]. As a result, the CO consumption was higher than the theoretical stoichiometry (~5.5 instead of 4 CO per acetate)[20]. Similar to batch cultivations, CO-1 produced $H_2$ and $CO_2$ through the water gas shift reaction in chemostats.

Next, we established cultures at $D = 0.20\,h^{-1}$ on CO to compare the behavior of CO-1 at two different growth speeds. The gas-liquid mass transfer rate was adjusted by increasing the stirrer speed to maintain biomass and acetate titers. Compared to $D = 0.10\,h^{-1}$, a higher specific CO uptake rate ($122.2 \pm 18.9\,mmol\,g^{-1}\,h^{-1}$) and a higher biomass yield ($1.64\,g\,mmol^{-1}$) were observed with a similar acetate yield ($0.14\,mol\,g^{-1}$), showcasing the potential of strain CO-1 for bioprocessing. Finally, increasing the dilution rate in a dynamic shift experiment until $0.25\,h^{-1}$ was possible (with wash-out observed at ~0.27 $h^{-1}$), thereby quantifying the maximum growth rate of the strain on CO.

**Steady-state transcriptomics analysis illuminates metabolic changes enabling carboxydotrophic growth**

Samples for transcriptomic analyses were collected from steady-state cultures at $D = 0.10\,h^{-1}$ (Table 2). Gene expression was quantified, and differential expression analysis was used to gain insights between strains and carbon/energy sources (Supplementary Data 2). Principal component analysis indicated that the choice of the strain had the strongest impact on expression variance (Supplementary Fig. 2). Comparison of the G-1 ($H_2/CO_2$) and the CO-1 ($H_2/CO_2$ or CO) datasets showed sporulation genes to be highly upregulated in CO-1 (19 genes with $log_2(FC) > 1$ and adj. $p$-value < 0.05), suggesting that CO-1 might highjack part of the sporulation signaling cascade to cope with CO despite the inability of *T. kivui* to form mature spores[37].

The strong carboxydotrophic phenotype exhibited by CO-1 could very well be the result of expression changes directly affecting the WLP, as most of the corresponding genes are borne by the megatransposon $Tn_{CO-1}$. Therefore, we investigated the strain-specific regulation of genes involved in CO metabolism. We compared both the CO-1 (CO) and G-1 ($H_2/CO_2$) data (Fig. 6 and Supplementary Data 2) and the CO-1 (syngas) and G-1 (syngas) data (Supplementary Fig. 3 and

**Table 2 | Data from steady-state chemostat cultivations of G-1 and CO-1 grown on mineral medium using $H_2/CO_2$, syngas or CO**

| Strain | G-1 | G-1 | CO-1 | CO-1 | CO-1 | CO-1 |
|---|---|---|---|---|---|---|
| Gas | $H_2/CO_2$ | Syngas | $H_2/CO_2$ | Syngas | CO | CO |
| D [h⁻¹] | 0.10 | 0.10 | 0.10 | 0.10 | 0.10 | 0.20 |
| Replicate # | 3 | 3 | 3ᵃ | 3 | 3 | 2 |
| Biomass [g L⁻¹] | 0.07 ± 0.01 | 0.30 ± 0.02 | 0.10 ± 0.01 | 0.29 ± 0.02 | 0.24 ± 0.02 | 0.26 ± 0.01 |
| Acetate [g L⁻¹] | 1.60 ± 0.18 | 6.12 ± 0.07 | 2.32 ± 0.14 | 6.01 ± 0.09 | 2.18 ± 0.09 | 2.12 ± 0.13 |
| $q_{H2}$ [mmol g⁻¹ h⁻¹] | 172.4 ± 45.6 | 97.6 ± 1.5 | 175.2 ± 14.0 | 92.9 ± 6.0 | − 16.7 ± 3.9 | − 26.1 ± 0.4 |
| $q_{CO2}$ [mmol g⁻¹ h⁻¹] | 80.3 ± 22.0 | 21.4 ± 4.2 | 89.2 ± 7.2 | 20.7 ± 1.0 | − 47.0 ± 6.0 | − 75.0 ± 0.7 |
| $q_{CO}$ [mmol g⁻¹ h⁻¹] | – | 52.3 ± 6.6 | – | 52.2 ± 5.7 | 80.0 ± 8.9 | 122.2 ± 18.9 |
| $q_{Ace}$ [mmol g⁻¹ h⁻¹] | 40.8 ± 10.6 | 34.3 ± 2.3 | 40.8 ± 1.3 | 34.3 ± 2.9 | 14.6 ± 0.5 | 27.3 ± 0.5 |
| $Y_{Biomass/S}$ [g mol⁻¹] | 0.60 ± 0.14 | – | 0.57 ± 0.05 | – | 1.26 ± 0.14 | 1.66 ± 0.26 |
| $Y_{Ace/Biomass}$ [mol g⁻¹] | 0.41 ± 0.11 | 0.34 ± 0.02 | 0.41 ± 0.01 | 0.34 ± 0.03 | 0.15 ± 0.01 | 0.14 ± 0.01 |
| C-balance [%] | 108.5 ± 3.1 | 99.2 ± 4.9 | 104.8 ± 8.0 | 95.8 ± 4.0 | 102.4 ± 4.1 | 109.4 ± 1.0 |

Values shown are average ± standard deviation. Source data are provided as a Source Data file.
ᵃNo biomass data available for replicate 3; biomass, specific rates, yields, and C-balance were calculated from two replicates.

Supplementary Data 2). In both cases, transcriptomics analysis indicated a potential reorganization of acetogen metabolism in CO-1, with expression of many redox-related genes being significantly regulated. The NAD⁺-dependent electron bifurcating NADPH:Fd oxidoreductase Nfn, and the Ech2 complex were significantly upregulated, with Ech2 exhibiting the highest changes (log₂(FC) between 1.33 and 2.52, CO versus $H_2/CO_2$ dataset). Ech2 as well as Nfn both use reduced ferredoxin (Fd²⁻) suggesting that the cellular balance between reduced and oxidized ferredoxin (Fd) might be critical for carboxydotrophy.

The monofunctional CO dehydrogenase CODH genes were counter-intuitively downregulated in CO-1 (log₂(FC) of − 1.96 and − 2.45 for *cooS* and *cooF1*, CO versus $H_2/CO_2$ dataset). Indeed, this result is unexpected as CODH is essential for CO metabolism[18]. Similarly, the acetyl-CoA synthase Acs genes were also slightly repressed, although Acs and CODH are the only enzymes directly utilizing CO. Similar to Ech2 and Nfn, both enzymes are linked to the cellular Fd²⁻/Fd pool. Reduced expression of CODH, an enzyme with high turnover rates[18], is expected to strongly limit CO oxidation and, therefore, the overgeneration of Fd²⁻. This synergizes with increased Ech2 activity, as Ech2 consumes Fd²⁻. In this view, maximizing regeneration of Fd would be linked with the carboxydotrophic phenotype, in a similar fashion that maximizing NAD⁺ regeneration favors efficient glucose uptake in glycolytic microorganisms.

### Overexpression of Ech2 is sufficient to promote carboxydotrophic growth in wild-type *T. kivui*

In our transcriptomics dataset, Ech2 upregulation in the CO-1 strain was particularly strong, and in the context of its inhibition by CO and its putative role in balancing the Fd pool, we hypothesized that the Ech2 expression level might be crucial for CO adaptation. The mechanism underlying the upregulation of Ech2 in the CO-1 strain is likely linked with the reorganization of the WLP genes on Tn_CO-1 (Fig. 7a). Indeed, the strongly expressed *acs* genes are interrupted at the level of *acsC* (TKV_RS09660) in Tn_CO-1, thereby removing the putative terminator located downstream of *gcvH* (TKV_RS09645). This regulatory element (identified with ARNold[38]) should terminate transcription of the entire WLP locus (organized in a single operon starting with *fhs*, TKV_RS09705, based on transcriptomic data). Consequently, in Tn_CO-1 the *ech2* genes are likely both under the control of their native promoter as well as the significantly stronger WLP promoter (about 11.5-fold stronger based on G-1 transcriptomic data).

We therefore sought to replicate a genetic organization similar to Tn_CO-1 by knocking in all *ech2* genes (TKV_RS09580-9615), as well as the hypothetical CDS upstream of these genes (TKV_RS12155, also significantly upregulated, log₂(FC) = 2.50), in a wild-type background.

We chose the strong, constitutive S-layer promoter[39], and a gene editing system based on the selective/counter-selective auxotrophic marker *pyrE* (conferring uracil prototrophy) and linear DNA transformation as previously described[34] (Fig. 7b). A 10.8 kb PCR fragment designed to knock in *pyrE* and *ech2* genes at the *pyrE* locus was used to transform a Δ*pyrE* strain derived from the wild type (Fig. 7c). Selection without uracil coupled to random isolation on plates yielded multiple strains displaying correct integration of *pyrE* but the loss of all or most of the cargo (Supplementary Fig. 4). The same cultures used for isolation were subcultured, using 50% CO as a selective pressure. After prolonged incubation at 66 °C, an increase of OD₆₆₀ was observed for the Ech2 knock-in transformations, but not for the negative control (knock-in of *pyrE* without any additional cargo). Further subculturing showed the ability of the culture to grow on CO as the sole carbon and energy source, and individual clones were isolated on plates and screened for carboxydotrophy. An isolate, further referred to as Ech2_KI, was shown to grow with 50% CO (2 bar) (Fig. 7c), albeit to a slightly lower OD₆₆₀ than CO-1 - possibly as a result of a lower growth rate than CO-1, as the experiment was time-limited. Ech2_KI was sequenced with both long- and short-read technologies, which revealed correct integration of the cargo at the *pyrE* locus, in contrast to the direct isolation step. This indicates that correct integration of the Ech2 knock-in was likely selected for by CO exposure, further demonstrating the importance of Ech2 for carboxydotrophic growth.

### The Ech2_KI strain bears additional small- to large-scale mutations

Since the selection step on 50% CO required one month, we checked for other mutations in the Ech2_KI strain that could have contributed to establish robust carboxydotrophic growth. In particular, we hypothesized that strong overexpression of the membrane-bound Ech2 complex could impose a metabolic burden, and that the expression level of Ech2 subunit genes might therefore require refining.

SNV/InDel analysis identified 21 mutations (Supplementary Table 2). A handful of the newly acquired mutations could play a significant role in rewiring acetogen metabolism, including pleiotropic regulators such as CodY (A167V) or PhoU (A109P), both of which could modulate carbon metabolism[40,41]. The Nfn subunit NfnB was also significantly extended in Ech2_KI (463 AA instead of 407 AA). Importantly, the C-terminus of NfnB is involved in NADPH binding[42], and the extension might, therefore, affect the redox activity of the whole complex.

In addition to SNV/InDel analysis, we performed de novo assembly of the Ech2_KI genome and found significant medium- and large-scale rearrangements, extensively remodeling the genomic landscape

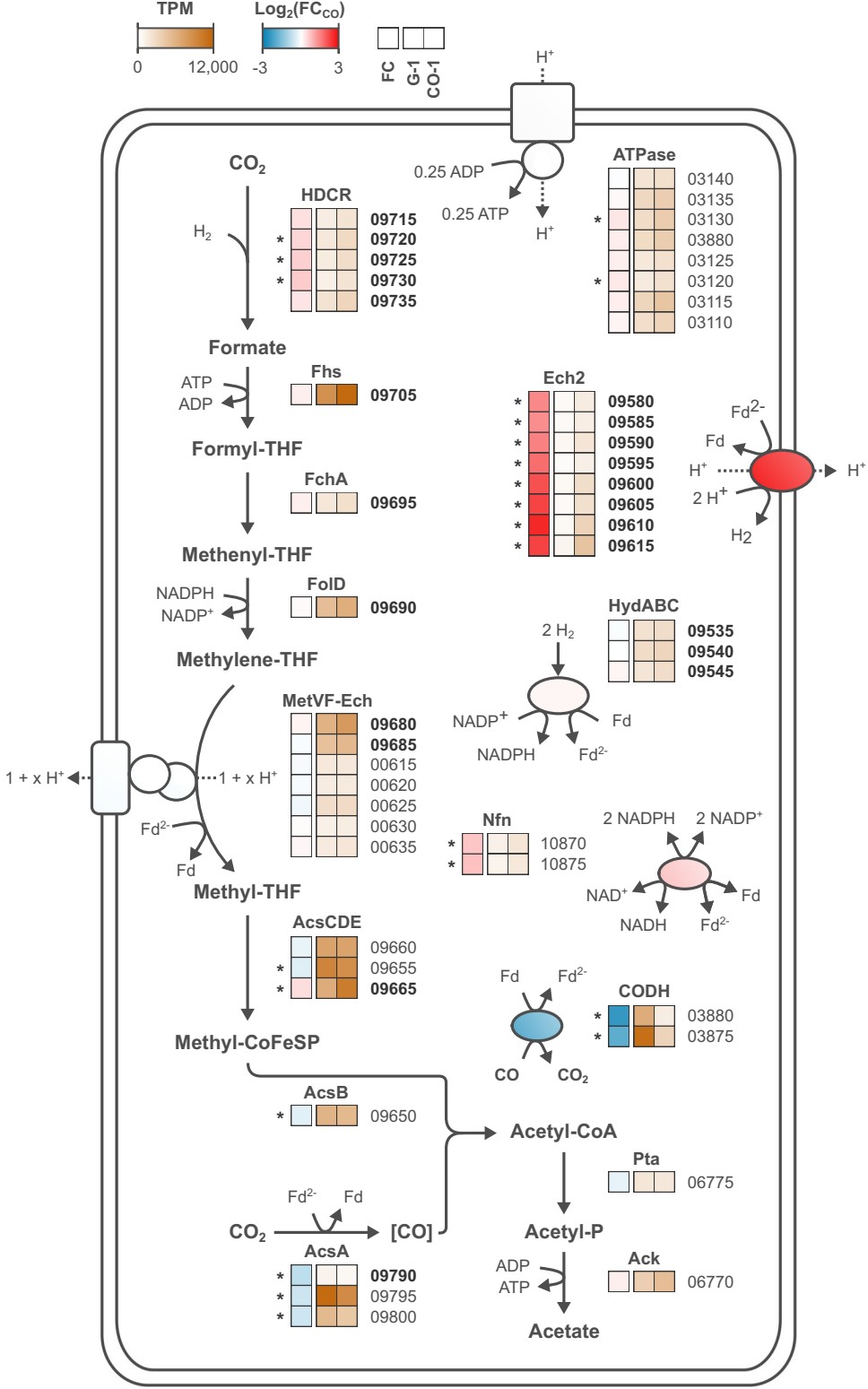

**Fig. 6 | Steady-state transcriptomics view of metabolism in G-1 and CO-1 strains.** Central metabolism comprising the WLP, as well as genes involved in energy conservation are shown. Mean TPM values as well as fold changes from the differential expression analysis (DESeq2) for G-1 ($H_2/CO_2$) and CO-1 (CO) are depicted as a heat map. Abbreviated locus tags (from the DSM 2030 genome) in bold indicate genes present on $Tn_{CO-1}$. Asterisks correspond to statistically significant values (two-sided Wald test, Benjamini and Hochberg adjusted $p$-value < 0.05). RNA-seq was performed using RNA extracted from triplicate continuous fermentation experiments described in Table 2. Source data are provided as a Source Data file.

(Supplementary Table 2 and Fig. 7d). Although $Tn_{CO-1}$ was not found in Ech2$_{KI}$, high transposon activity resulted in the insertion or excision at multiple (6) loci of single gene transposases, locally affecting several genes. Most of this transposon activity was mediated by an IS*Lre2*-family transposase, i.e., a different transposase as the one involved in the formation of the megatransposon $Tn_{CO-1}$ in CO-1. Transposon excision notably occurred in close proximity to the Nfn encoding genes (upstream of TKV_RS10865, encoding a secondary alcohol

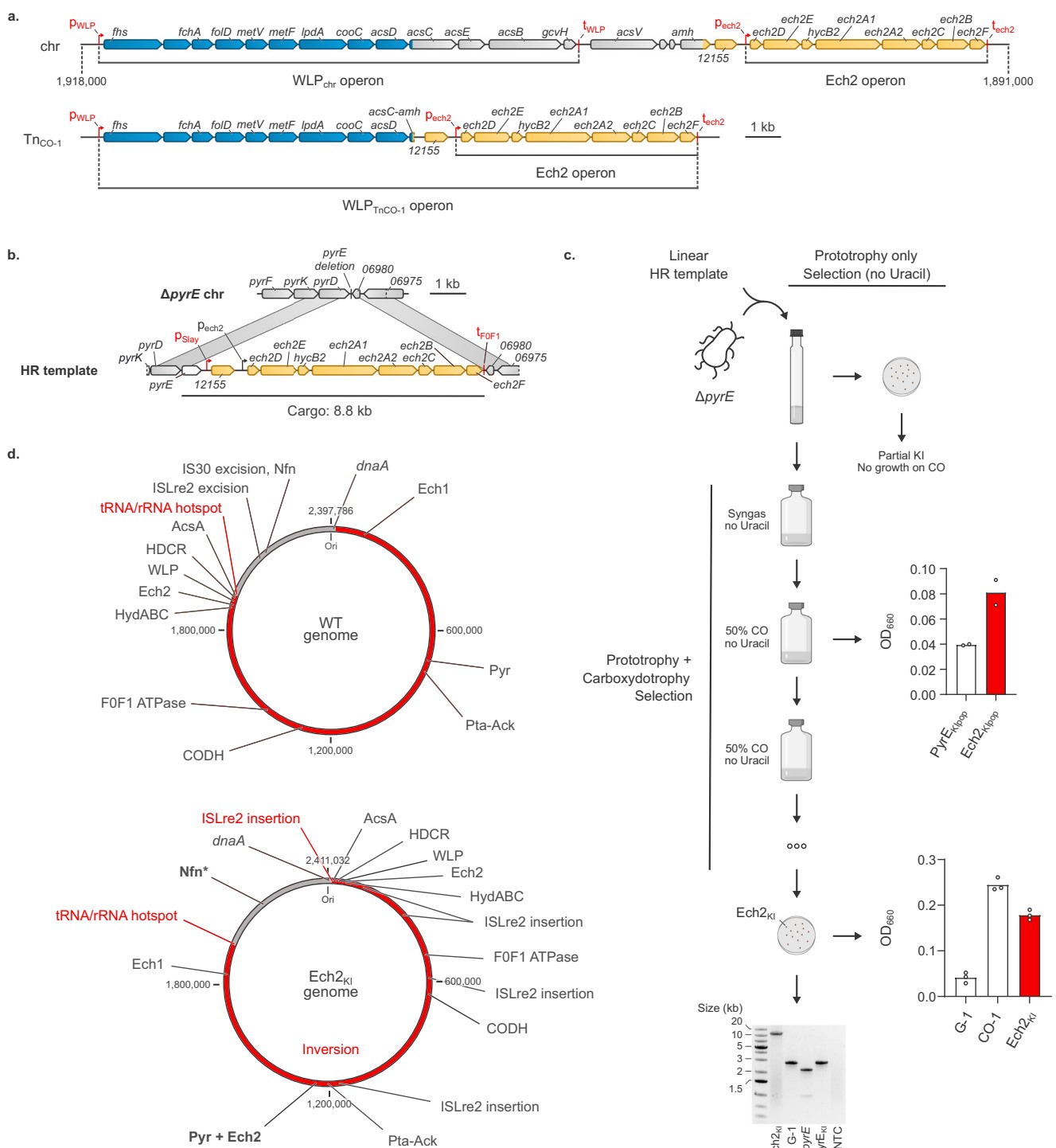

**Fig. 7 | The Ech2 enzyme is crucial for CO adaptation in *T. kivui*. a** Overview of the WLP and Ech2 locus organization on the chromosome and Tn$_{CO-1}$. **b** Genome editing design for the overexpression of Ech2. **c** Genome editing pipeline: direct selection of prototrophic cells, versus coupled selection of prototrophy and carboxydotrophy. The top bar plot shows OD$_{660}$ after 1 month of selection without uracil at 66 °C with 50% CO (2 bar). The bottom bar plot shows the carboxydotrophic growth of strain Ech2$_{KI}$ after 3 days in triplicate serum bottle fermentations with 50% CO in mineral medium, starting from OD$_{660}$ = 0.03. NTC: no template control. The PCR genotyping of Ech2$_{KI}$ was performed as a single experiment. **d** Large- and medium-scale rearrangements in the carboxydotrophic Ech2$_{KI}$ strain. The red segment corresponds to a large inversion event. WT genome refers to isolate G-1 (GCA_963971585). Bold: mutated genes. Except for *dnaA*, names featured in (**d**) refer to loci and not genes. Source data are provided as a Source Data file.

dehydrogenase), which might affect *nfnAB* expression. Whole genome alignment further revealed the inversion of a 1937 kbp fragment (approximately 80% of the wild-type genome). This inversion is flanked on one side by an IS*Lre2*-family transposase, and on the other side by the same tRNA/rRNA hotspot identified at the end of the CO-1

duplication (Fig. 1a). The inversion completely reshapes the genome (Fig. 7d), and notably 1) brings the WLP locus much closer to the origin of replication (at about 15 kbp from the start) and 2) pushes the knock-in locus away from the origin of replication (at about 1150 kbp, halfway through the genome). In bacteria, the position of a gene relative to the

origin of replication strongly affects its expression, with genes closer to the origin of replication being, on average, more expressed[43]. In Ech2$_{KI}$, the novel positional bias resulting from the inversion could, therefore, potentially increase the expression of all genes from the WLP locus, while decreasing the expression of the Ech2 knock-in.

In order to verify the effect of these rearrangements on gene expression, we cultivated strains PyrE$_{KI}$ (control strain with insertion of *pyrE*), CO-1, and Ech2$_{KI}$ in serum bottles with syngas, and compared their relative gene expression levels using transcriptomics (Supplementary Fig. 5 and Supplementary Data 3). While genes of the WLP were not differentially expressed, the genes introduced in Ech2$_{KI}$ (TKV_RS12155 as well as the *ech2* genes) were highly upregulated (log$_2$(FC) comprised between 3.09 and 4.01, all in the top differentially expressed genes in the Ech2$_{KI}$ versus PyrE$_{KI}$ comparison). The level of expression of Ech2 subunit genes in Ech2$_{KI}$ is, however, about 5-10-fold lower (in TPM) than that of the S-layer protein (TKV_RS11315), whose promoter was used to drive the expression of the *ech2* subunit genes. Therefore, the large chromosomal rearrangement in Ech2$_{KI}$ could indeed have resulted in a lower expression of these genes, possibly as a means to decrease the metabolic burden linked with high Ech2 expression while still allowing for carboxydotrophic growth.

## Discussion

In this work, we adapted *T. kivui* to grow solely on CO and rapidly obtained the CO-1 strain within 31 generations which tolerates and uses CO efficiently. Growth on CO was shown to be particularly fast ($\mu_{max}$ = 0.25 h$^{-1}$, T$_d$ ~ 2.8 h), in particular compared to values reported previously for CO-adapted *T. kivui* (0.017 h$^{-1}$, T$_d$ ~ 40.8 h)[8]. This stark difference is perhaps linked to the adaptation procedure (yeast extract was omitted in our work), resulting in a harsher selection pressure toward a highly potent carboxydotrophic strain. To the best of our knowledge, the growth rate of CO-1 is higher than the growth rate of any other acetogen grown on CO in a chemically defined medium. The closest $\mu_{max}$ was reported for *C. autoethanogenum* in chemostat cultivations (0.12 h$^{-1}$, T$_d$ ~ 5.8 h)[44], highlighting the relevance of our phenotype for functional analysis and for bioprocessing of syngas. *T. kivui*, as a promising biocatalyst for syngas conversion, is characterized by co-consumption of CO and H$_2$ in batch cultures and CO, H$_2$, and CO$_2$ co-consumption in chemostats with high catalytic activity as demonstrated by the high specific growth rates and gas uptake rates, indicating that syngas conversion at high, industrially relevant space-time-yields is possible. We further showed that the strong carboxydotrophic ability of CO-1 is conferred by the mobilization of the CO-dependent extrachromosomal megatransposon Tn$_{CO-1}$. Indeed, this mobile element is essential to mediate the CO-1 phenotype, as loss of Tn$_{CO-1}$ results in complete loss of carboxydotrophy in CO-1.

Tn$_{CO-1}$ surprisingly rearranges two genomic loci into a single circular DNA molecule. Tn$_{CO-1}$ bears only one transposase gene, i.e., the one present in the IS*1182*-like element (as predicted by ISfinder[45]). Although little information is available for these mobile elements, IS*1182*-related transposases seemingly operate via distinct mechanisms[46–49]. They can drive transposition events with or without passenger genes, with or without duplication of the target insertion site, with or without deletion at the target insertion site, with recognition of a target sequence or alternatively a target DNA structure. However, a major common point is that transposition depends on inverted repeat sequences (IR) as recognition sequences for mobilization. Using ISfinder[45], we found a closely related IS*1182* transposon annotated in *Caldanaerobacter subterraneus* subsp. *tengcongensis*. The imperfect IRL and IRR (IR left and right) matched very well with sequences found directly upstream and downstream *T. kivui* IS*1182*-like transposase CDS. Furi et al. showed that an IS*1182*-related transposase was able to highjack a passenger gene (in that case, a gene involved in biocide resistance) via a degenerate IRL sequence located upstream of that passenger gene[46]. When we examined the ends of

each duplicated chromosomal region in CO-1, we found one such sequence at the end of the second duplication (referred to as IRR', Supplementary Fig. 6). IS*1182*-mediated transposition using IRL and IRR' as recognition motifs would typically result in a 94 kb circular intermediate if transposition follows a classical copy-out-paste-in process[50]. We could not, however, find such IR sequences at the P3/P4 junction, and how exactly the corresponding region is excised in Tn$_{CO-1}$ remains elusive. Regardless of the activation and formation mechanisms, Tn$_{CO-1}$ mobilization turned out to be remarkably stable in CO-1, as it could be consistently initiated by the addition of CO.

In acetogens, inhibition of [FeFe]-hydrogenases is thought to be the primary driver of CO toxicity[11,14,21]. Here, we characterized an exceptionally robust carboxydotrophic acetogen, in which the major [FeFe]-hydrogenases HydABC and HDCR where neither mutated nor significantly up- or downregulated (Fig. 6). In contrast, the Ech2 [NiFe]-hydrogenase was significantly upregulated. Ech2 was previously shown to be inhibited by CO[12], however the IC$_{50}$ value (~ 200 μM, compared to ~ 0.25 μM for the HDCR) suggests Ech2 inhibition does not occur in vivo, as the physiological intracellular CO concentrations are expected to be below the μM range[51] and given that CODH activity was likely to be sufficient to completely oxidize CO diffusing across the cell membrane[52]. In addition, previous work showed that an Ech2 knock-out prevented adaptation of *T. kivui* to CO + yeast extract[21], while our study demonstrated that *ech2* overexpression (i.e., as in the Ech2$_{KI}$ strain) could kick-start growth on CO as sole energy/carbon source. Ech2 provides a key activity as it allows the regeneration of Fd. As the Fd$^{2-}$-generating CODH is also strongly downregulated in CO-1, the differential expression of both enzymes likely synergizes by balancing Fd$^{2-}$ generation and consumption (to maintain redox balance). In the wild type, the accumulation of Fd$^{2-}$ through CO oxidation and simultaneous depletion of the Fd pool is very likely detrimental for central metabolic reactions. Indeed, critical enzymes for redox balance, such as HydABC and Nfn, require Fd and Fd depletion might, therefore, effectively paralyze metabolism.

During the final stages of this study, Baum et al. published Illumina data of CO-adapted *T. kivui* strains[21]. SNP/InDel analysis suggested CO adaptation derived from a mutation arising in the HDCR complex. By performing coverage analysis, we uncovered an unreported ~125 kb duplication in 4 out of 5 analyzed clones, between position ~ 125 kb and 250 kb of the published genome of *T. kivui* DSM 2030 (NZ_CP009170)[53]. In this region, average coverage is ~ 3.4-fold compared to the rest of the genome and interestingly stops at a IS*Lre2*-type transposase gene (TKV_RS01200, the same involved in Ech2$_{KI}$ inversion), suggesting that another megatransposon might be at play in CO adaptation of the corresponding clones. Altogether, many genes are duplicated, including some that might be highly relevant for carboxydotrophic growth, such as those encoding the energy-converting hydrogenase complex Ech1 subunits (TKV_RS00615-0655), and gene expression analysis showed upregulation of Ech1 subunit genes. However, the role of Ech1 in *T. kivui* metabolism is not fully understood, and so far, no Fd$^{2-}$ oxidation activity has been determined for Ech1[21]. In contrast, the upregulation of Ech2 genes also found in these strains might suggest an underlying CO tolerance mechanism at least partly similar to the one in CO-1, where balancing the Fd$^{2-}$/Fd pool appears to be crucial.

Efficient consumption of excess Fd$^{2-}$ by Ech2 under highly reduced CO fermentation is, therefore, likely the key to promote carboxydotrophy in *T. kivui*, provided their expression is sufficient to balance CODH-mediated ferredoxin reduction. Indeed, Ech2-mediated Fd$^{2-}$ regeneration can serve as a redox pressure relief valve yielding H$_2$, which can escape the cell (which is supported by our experiments, as we observed H$_2$ production from CO). In addition, carbon reduction in the methyl branch of the WLP could be an alternative route for Fd$^{2-}$ reoxidation as Fd$^{2-}$-dependent methylene-THF reductase activity has been demonstrated[19,20].

In the natural carboxydotrophic acetogen *Clostridium autoethanogenum*, a CO tolerance mechanism in principle similar to the one we describe for *T. kivui* has been proposed. Indeed, the ethanol fermentation pathway via the aldehyde:ferredoxin oxidoreductase (AOR) of *C. autoethanogenum* plays a crucial role to maintain a balance between Fd and Fd$^{2-}$ by consuming Fd$^{2-}$ under high CO concentration, using acetate as an electron sink[52,54–56]. In *Eubacterium limosum*, adaptation to CO resulted in mutations either affecting the CODH subunit of the acetyl-CoA synthase (AcsA or cognate maturation protein CooC2)[9] or the beta subunit (AcsB)[57]. In the absence of a dedicated monofunctional CODH, AcsA is responsible for CO oxidation in *E. limosum*[58]. The mutation reported for AcsB was suggested to facilitate a closed conformation of the ACS/CODH complex, improving its activity in the direction of $CO_2$ reduction, Fd$^{2-}$ oxidation, and acetyl-CoA formation[57,59]. Therefore, it is likely that AcsA and AcsB mutations both negatively affect the rate of CO oxidation and consequently of Fd$^{2-}$ generation. As a result, adaptation of *E. limosum* to CO could follow the same rules as in *T. kivui* CO-1. In *A. woodii*, recent results have shown that mixotrophy on CO + yeast extract is possible when the HDCR is adapted from an $H_2$-consuming enzyme to a Fd$^{2-}$-consuming enzyme[60], suggesting fine-tuning the redox balance could also be at play. In that case, circumventing the CO inhibition occurring at the HDCR level might be equally or more important, as the Fd$^{2-}$ activity of the HDCR is not CO-inhibited[13].

Thermophiles in general and *Thermoanaerobacter* species, in particular, tend to display low genetic barriers with, e.g., highly efficient natural competency[61], favoring in nature horizontal gene transfer (HGT). Thermophilic bacteria live in extreme environments, which requires fast adaptive capability[62]. Interestingly, *T. kivui* is the only *Thermoanaerobacter* described so far capable of autotrophic growth. Because most genes involved in the WLP are colocalizing in the chromosome of *T. kivui*[53], the acquisition of these genes was hypothesized to stem from a single large HGT event[22], suggesting that autotrophy could be fully transferrable through horizontal exchange. That view was recently challenged, as a phylogenetic analysis suggested autotrophy was vertically inherited, and lost in all *Thermoanaerobacter* spp. but *T. kivui*[63]. Throughout our study, we witnessed two cases of mobility of the WLP locus: first through the mobilization of Tn$_{CO-1}$, and second via a large inversion of the genome. Both events were likely mediated by transposases (from the IS*1182* and IS*Lre2* families, respectively), as these were found in close proximity of the corresponding rearrangements. In both cases, a tRNA/rRNA hotspot (TKV_RS09805-9855) marks the other end of the rearrangement. tRNA and rRNA genes are strategic targets for transposition elements, as their sequences are highly conserved among taxa, favoring cross-species transfer of these mobile elements[25–33]. Collectively, the high mobility of the DNA segment bearing the WLP genes, the involvement of transposases, and the presence of a tRNA/rRNA hotspot all support the hypothesis that *T. kivui* acquired autotrophy via horizontal gene transfer, adding yet another example of the fundamental role transposons have played and keep on playing in the evolution of life[64,65].

## Methods

### Strains, medium and culture conditions
Supplementary Data 4 contains the strains used in this study. The *T. kivui* strains were cultivated at 66 °C or 55 °C on mineral medium containing per liter: 7.80 g $Na_2HPO_4 \times 2 H_2O$, 6.90 g $NaH_2PO_4 \times H_2O$, 0.21 g $K_2HPO_4$, 0.16 g $KH_2PO_4$, 0.25 g $NH_4Cl$, 0.225 g $(NH_4)_2SO_4$, 0.448 g NaCl, 0.121 g $MgSO_4 \times 7 H_2O$, 0.07 g $CaCl_2 \times 2 H_2O$, 1 mL trace element solution (based on DSMZ141). The trace element solution contained per liter: 15.00 g nitrilotriacetic acid, 5.00 g $MnSO_4$ x $H_2O$, 1.50 g $CoCl_2$ x 6 $H_2O$, 1.80 g $ZnSO_4$ x 7 $H_2O$, 0.10 g $CuSO_4 \times 5 H_2O$, 0.10 g $H_3BO_3$, 0.10 g $Na_2MoO_4 \times 2 H_2O$, 0.33 g $NiSO_4 \times 6 H_2O$, 0.003 g $Na_2SeO_3 \times 5 H_2O$, 0.004 g $Na_2WO_4 \times 2 H_2O$. $FeSO_4 \times 7 H_2O$ was added to a final concentration of 3 mg L$^{-1}$ for solid media, serum bottles, and batch bioreactors and 28.6 mg L$^{-1}$ in the chemostat feed medium. For batch bioreactor and chemostat feed medium, the $NH_4Cl$ concentration was increased to 1 g L$^{-1}$.

Glucose (5 g L$^{-1}$) or gas was used as a carbon/energy source as necessary in serum bottles or Hungate tubes sealed with rubber stoppers. For glucose fermentation, 2 bar $N_2/CO_2$ (80:20) was used as make-up gas. For gas fermentation, the headspace of serum bottles was flushed with the appropriate gas, and the pressure was set to 2 bar unless otherwise stated. Gasses for the serum bottles were mixed with Brooks 4800 series mass flow controllers (Brooks Instrument, Hatfield, USA), except for CO, which was added separately, and the concentration set by a fraction of pressure. The medium was supplemented with 2 g L$^{-1}$ yeast extract (for cultivating the Δ*pyrE* strain only), and with 200 mg L$^{-1}$ kanamycin/neomycin or 100 mg L$^{-1}$ 5-fluoroorotic acid when appropriate.

For the solid medium, the mineral medium was supplemented with 7 g L$^{-1}$ gelrite, 5 g L$^{-1}$ glucose, and 2 g L$^{-1}$ yeast extract (except for uracil prototrophy selection). 200 mg L$^{-1}$ neomycin was added when appropriate. Cells were grown in custom-made metallic jars pressurized at 2 bar $N_2/CO_2$ (80:20) at 55 °C or 66 °C until colony formation.

*E. coli* DH10B (Thermo Fisher, MA, USA) or CopyCutter EPI400 (Epicenter, WI, USA) were cultivated on LB medium (10 g L$^{-1}$ tryptone, 10 g L$^{-1}$ NaCl, 5 g L$^{-1}$ yeast extract) with 50 mg L$^{-1}$ kanamycin, 100 mg L$^{-1}$ carbenicillin and 1X CopyCutter induction solution when appropriate. For solid medium, 15 g L$^{-1}$ agar was added.

### Adaptation to CO
*T. kivui* DSM 2030 was obtained from the DSMZ collection and directly used for CO adaptation without reisolation. CO adaptation was undertaken by serially growing *T. kivui* in 125 mL serum bottles filled with 20 mL mineral medium and 2 bars of the appropriate gas. Cells were grown at 66 °C with shaking in a water bath (Memmert, Schwabach, Germany). Upon visible growth, 1 mL of cell culture was transferred to the next serum bottle for every adaptation step, until high growth on 100% CO was observed (OD$_{600}$ = 0.2). For clonal strain isolation, cells were streaked on plates under heterotrophic conditions, and isolated colonies were grown in Hungate tubes with the appropriate gas mixture at 2 bar.

### Fermentation in bioreactor
Bioreactor cultivations were carried out in a DASBox® Mini Bioreactor system (Eppendorf AG, Hamburg, Germany). Batch cultivations were conducted with a working volume of 250 mL and an agitation rate of 600 rpm for the growth on syngas ($H_2/CO_2/CO/N_2$: 24/21/52/3) and an agitation of 300 rpm for the growth on 100% CO. An aeration rate of 0.05 vvm was applied.

Continuous cultivations were carried out with a working volume of 208 mL. Agitation was adjusted to maintain a constant acetate titer for the growth on $H_2/CO_2$ ($H_2/CO_2/N_2$: 58/9/33), syngas ($H_2/CO_2/CO/N_2$: 58/9/30/3), and CO (CO/N$_2$: 52/48) with an aeration rate of 0.06 vvm. All gas mixtures were premixed and obtained from Messer (Messer Austria GmbH, Gumpoldskirchen, Austria).

All bioreactor cultivations were operated at a constant temperature of 66 °C using a static water bath. The pH was set to 6.4 and constantly monitored with an Easy Ferm Plus K8/120 pH electrode (Hamilton, Reno, NV, USA) and automatically adjusted by the addition of 5 M KOH using an MP8 multi-pump module (Eppendorf AG, Hamburg, Germany). Uniform gassing was ensured using a sintered microsparger with a defined pore size of 10 μm (Sartorius Stedim Biotech GmbH, Göttingen, Germany). Prior to inoculation, the reactor was flushed with the appropriate gas mixture for at least 3 h to ensure anaerobic conditions. The bioreactors were inoculated to an initial OD$_{600}$ of 0.1. Chemostat cultivations were switched to continuous mode after the pH regulation was activated for the first time, using a dilution rate of 0.1 h$^{-1}$ (subsequently increasing to 0.2 h$^{-1}$ for CO-1 on CO).

Reactor sampling was carried out every 2.5 h in batch mode and once per day for continuous mode, monitoring $OD_{600}$ (ONDA V-10 Plus Visible Spectrophotometer) and product content by HPLC (Ultimate 3000 High-Performance Liquid Chromatograph, Thermo Fisher Scientific, Waltham/MA, USA). For HPLC, an Aminex HPX-87H column ($300 \times 7.8$ mm, Bio-Rad, Hercules/CA, USA) was used to analyze the samples (mobile phase 4 mM $H_2SO_4$, 0.6 mL min$^{-1}$, column heated at 60 °C), using a refractive index (Refractomax 520, Thermo Fisher Scientific, Waltham, MA, USA) and a diode array detector (Ultimate 3000, Thermo Fisher Scientific, Waltham, MA, USA) for quantification.

For dry cell weight determination, 50 mL of culture broth was collected at the end of the batch cultivation and at steady-state conditions (constant values for $OD_{600}$, acetate titer, and gas uptake/production rates for 4, 5 volume changes) for the continuous cultivation. The tubes were centrifuged and washed two times with 0.9% (w/v) NaCl. The biomass sample was then suspended with 5 mL distilled water, transferred into pre-weighed glass tubes, dried out for 24 h at 120 °C, and cooled down in a desiccator for 1 h before weighing. Biomass determination was performed in triplicates.

A Micro GC Fusion® Gas Analyzer (Inficon Holding AG, Bad Ragaz, Switzerland) was used to analyze the off-gas of all reactors every hour using a Valco stream selector from VICI® (Valco® Instruments Co. Inc., Houston, Texas). To account for the volume change in the off-gas, the $N_2$ concentration was used for the normalization of the gas composition. A Rt-Molsieve 5 A column using argon as carrier gas was used to detect $N_2$, $H_2$, and CO. A second Rt-Q-Bond was used to detect $CO_2$ with helium as carrier gas. A thermal conductivity detector was used for quantitative analysis. Online GC values were used to calculate gas uptake rates[66].

## Short-read genome sequencing

For strain resequencing (CO−1, CO-2, CO-3), cell pellets were harvested (5–10 OD units, 16,000 g for 10 min at 4 °C) from growing cultures, frozen at − 80 °C and sent to Microsynth GmbH (Balgach, Switzerland). Cells were subsequently lysed with lysozyme overnight in STET buffer (10 mM Tris-HCl, 1 mM EDTA, 100 mM NaCl, and 5% Triton X-100) at 37 °C, followed by proteinase K/RNase A treatment. DNA was isolated from the supernatant using the DNeasy kit (Qiagen) according to the manufacturer's instructions. Illumina's DNA Prep tagmentation library preparation kit was used according to the manufacturer's recommendations to construct the libraries. Subsequently, the Illumina NovaSeq platform with an SP flow cell and a 200 cycles kit were used to sequence the libraries ($2 \times 75$ or $2 \times 150$ bp). Paired-end reads which passed Illumina's chastity filter were subject to de-multiplexing and trimming of Illumina adapter residuals using Illumina's bcl2fastq software version 2.20.0.422 (without further refinement or selection).

For polishing de novo assemblies obtained from long-read ONT sequencing, Illumina reads were obtained from the same DNA sequenced with ONT. Cells were treated with lysozyme in TE buffer, and subsequent DNA isolation was performed using the Quick-DNA Miniprep Plus kit (Zymo Research, Irvine, CA, USA) according to the manufacturer's instructions. CO-1 DNA was sequenced by Microsynth GmbH as described above, except the TruSeq DNA library kit was used. Ech2$_{KI}$ DNA library was prepared by PlasmidSaurus (Eugene, OR, USA) using the seqWell ExpressPlex 96 library prep kit according to the manufacturer's instructions and was further sequenced on a NextSeq2000 sequencer (paired-end, $2 \times 150$ bp).

## SNV/InDel analysis

For the analysis of the evolved strains, Illumina reads were mapped onto the wild-type assembly (G-1, OZ020628 [67]) with Geneious Prime 2024.0.5 (https://www.geneious.com) using the Minimap2[68] plugin (v2.24, short read mode, default parameters). SNV/InDel analysis was performed with the built-in function of Geneious, filtering for

mutations arising in more than 80% of the mapped reads for a given position. Mutated genes are reported with the locus tag imported from the DSM 2030 genome[53].

## Long-read genome sequencing

Cell pellets were harvested from growing cultures, pelleted, and directly processed for DNA extraction. Cells were treated with lysozyme in TE buffer, and subsequent DNA isolation was performed using the Quick-DNA Miniprep Plus kit (Zymo Research, Irvine, CA, USA) according to the manufacturer's instructions. Extracted DNA was sent to PlasmidSaurus for ONT sequencing. Libraries were prepared with the Rapid Barcoding Kit 96 V14 (SQK-RBK114.96, ONT) according to the manufacturer's instructions, without DNA shearing or size selection, before subsequent sequencing (PromethION, R10.4.1 flowcell). Raw reads were obtained after basecalling, barcode splitting, and adapter trimming (Guppy v.6.4.6, super-high accuracy mode).

## De novo assembly

For the CO-1 strain, the 390,315 raw reads ($N_{50} = 5697$) were trimmed with PoreChop[69] v0.2.4, down-sampled to 100x coverage with Seqkit v2.4.0[70] and sorted for longest reads, resulting in 15,957 reads ($N_{50} = 15,344$). Draft assembly was performed with Canu v2.2[71] with parameters "genomeSize = 2.4 m maxThreads = 7 useGrid = false -nanopore", yielding 5 linear contigs. Based on the similarity to the DSM 2030 genome, the longest contig (2,424,656 bp) was identified as the CO-1 chromosome. Overlaps of both ends of this contig were determined by BLASTn[72], the chromosome was circularized based on these overlaps and rotated to start at *dnaA* gene with Seqkit v2.4.0[70]. The genome was polished and validated using the corresponding Illumina data as described above, resulting in a 2,396,992 bp chromosome. For the 4 remaining contigs (36,204 bp, 35,793 bp, 55,987 bp, and 60,479 bp), regions overlapping the chromosomal contig were found at the duplication sites previously determined through Illumina sequencing using BLASTn[72]. These 4 contigs significantly overlapped each other, which allowed the assembly of a single, circular extrachromosomal element (85,811 bp). The sequence was rotated to start at the IS*1182* element of the first duplication. Both sequences were annotated with a local instance of NCBI's Prokaryotic Genome Annotation Pipeline (PGAP)[73].

For the Ech2$_{KI}$ strain, de novo assembly and polishing were performed by PlasmidSaurus. The 135,120 reads ($N_{50} = 4200$) were filtered using Filtlong[74] (v0.2.1, default parameters) to exclude the 5% reads with the lowest average quality. Reads were down-sampled to 100 x coverage with Filtlong, weighted by quality score. De novo assembly was performed with Flye[75] (v2.9.1), with parameters selected for high-quality ONT reads. The resulting assembly was polished using the down-sampled ONT reads with Medaka[76] v1.8.0 and all Illumina reads with Polypolish[76] v0.6.0, yielding a 2,411,034 bp chromosome. The chromosome was rotated to start at the same position as wild-type *T. kivui* (OZ020628) and annotated using the PGAP[73].

## Structural variation analysis

CO-1 and Ech2$_{KI}$ chromosomes were aligned to the wild-type assembly using Mauve[77]. Transposon translocations and sequence inversion were manually inspected using Geneious Prime.

## Transposon quantification

Seqkit v2.4.0[70] was used to select raw ONT reads bearing Tn$_{CO-1}$-specific junctions sequences, defined as 50 bp sequences composed of 25 bp specific of each side flanking a given junction. P1/P2 junction sequence was 5'-aaaacagggcacttaagtgccctgttcagactgttgacaaaatatcggga-3', and P3/P4 junction was 5'-agtatatcttttttatattatccatattgcaaaagcaagacaagttggaa-3'. For quantification, the number of reads bearing each junction was normalized using sequencing depth (in Gb).

## Genotyping

PCR was routinely used to assess genotype by targeting chromosomal loci (*pyrE*, CO-1 specific deletion), SPF-B017, or $Tn_{CO-1}$ (at P1/P2 and P3/P4 junctions). S7 Fusion polymerase (Biozym, Hessisch Oldendorf, Germany) was used to amplify DNA from purified genomic DNA or directly from cell cultures. For semi-quantitative PCR, gDNA was quantified with Qubit (Thermo Fisher), diluted to $2\,ng\,\mu L^{-1}$ and used for PCR, limiting cycling to visualize DNA without saturation after gel electrophoresis ($1\,\mu L$ DNA in $50\,\mu L$ PCR mix, 28 cycles for PCR).

## Transcriptomics

Cells from steady-state chemostats or serum bottles (in log phase, $OD_{660}$ between 0.2 and 0.3) were pelleted (1 OD unit, $12,000\,g$ at $-4\,^{\circ}C$ for 5 min) and shock-frozen in liquid nitrogen. Pellets were next shipped on dry ice to Microsynth GmbH for processing. DNA/RNA Shield (Zymo Research, Irvine, CA, USA) was used to resuspend cell pellets, and cells were lysed by bead beating (using an MP Bio FastPrep with 0.1/0.5 mm Zirconia beads, 2 times 1 min at speed 5). The soluble fraction was thereafter used for RNA isolation using the RNeasy Plus 96 Kit (Qiagen), and RNA quality was checked on a Fragment Analyzer (Agilent). Libraries were prepared with the Stranded Total RNA Prep Kit with Ribo-Zero Plus Kit (Illumina), following the manufacturer's instructions. Sequencing was performed on an Illumina NovaSeq 6000 device ($2 \times 150\,bp$). The paired-end reads, which passed Illumina's chastity filter, were subject to de-multiplexing and trimming of Illumina adapter residuals using Illumina's bcl2fastq software version v2.20.0.422. RNA-seq reads were mapped onto *T. kivui* WT genome (GCA_963971585) using Geneious Prime 2024.0.5 with the built-in Geneious mapper (default parameters). After counting (Geneious, default parameters), the DESeq2[78] plugin for Geneious was used for PCA and differential expression analysis (default parameters). Genes (referred to with the locus tag imported from the DSM 2030 genome[53]) were considered differentially expressed when $|log_2(FC)| > 1$ and Benjamini and Hochberg adjusted *p*-value < 0.05.

## Cloning

The GoldenMOCS assembly system described before was used for all cloning[39]. An overview of plasmids and primers used in this study is given in Supplementary Data 4. The DH10B and CopyCutter (for large vectors) *E. coli* strains were transformed using classical chemically induced competence.

SPF-B017, a recipient BB3[79] *E. coli-T. kivui* shuttle vector was constructed by combining several parts by BbsI Golden Gate assembly, i.e., a BsaI-compatible AD linker, the pMU131 Gram-positive replicon from BB2_pMU131[39] (shortened to remove a BsaI site), the thermostable kanamycin resistance marker from pMU131[61], and a ColE1 Gram-negative origin of replication.

The $\Delta pyrE$ vector contained *pyrE* flanking regions (1000 bp each) so that deletion results in the removal of most of the *pyrE* CDS, except for the first three and last two codons. The resulting design was ordered as a gene synthesis (Twist Bioscience, CA, USA) cloned in a high-copy vector.

The $PyrE_{KI}$ plasmid was assembled in two steps. First, two homology arms upstream and downstream *pyrE* (1000 bp) were amplified separately with fusion sites A4 and 4D[80]. The upstream homology arm also contained the *pyrE* gene, and both amplicons were cloned as BB2 donors using the pMini2.0 kit (NEB, MA, USA). The resulting plasmids were cloned into an *E. coli* BB3 AD recipient vector[81] by Golden Gate assembly using BsaI-HFv2 (NEB).

The $Ech2_{KI}$ vector was similarly cloned in multiple steps. First, the *pyrE* downstream homology arm similar as the one described above – but with CD fusion sites[80] – was subcloned using the pMini2.0 kit (NEB). The AB vector containing the upstream homology arm, *pyrE* CDS, and the BBa_B1001 terminator was synthesized (Twist Bioscience). The BC vector containing *T. kivui* S-layer promoter[39], an

Esp3I 23 linker cassette, and the terminator from the F0F1 ATPase operon (downstream TKV_RS03140) was also synthesized. The AB, BC, and CD parts were assembled in SPF-B017 with Golden Gate assembly (BsaI), yielding BB3_pyrE_Esp3I. Finally, the Ech2 operon, including the upstream TKV_RS12155 hypothetical CDS, was amplified by PCR and cloned into BB3_pyrE_Esp3I by Golden Gate assembly using Esp3I (NEB), yielding the final ~15 kb $Ech2_{KI}$ vector.

## *T. kivui* transformation and genome editing

A replicative plasmid (SPF-B017, for transforming CO-1), non-replicative plasmid (*pyrE* knockout), or linear DNA (knock-in at *pyrE* locus) were used for transformation. To generate linear DNA, homology regions were amplified by PCR from a plasmid (see cloning section), and the residual plasmid was removed by DpnI digestion.

*T. kivui* was transformed exploiting the natural competence of the strain[39]. Briefly, 2 mL of medium were inoculated in a Hungate tube with $1\,\mu g$ DNA. Cells were grown overnight at $55\,^{\circ}C$ under 1 bar overpressure of $N_2/CO_2$ (80/20). The selection was performed by spreading onto plates ($\Delta pyrE$ with 5-fluoroorotic acid) or directly by subculturing into selective liquid medium (CO-1 SPF-B017 with neomycin, or in chemically defined medium without uracil for the $PyrE_{KI}$ and $Ech2_{KI}$ strains) and incubated at $55\,^{\circ}C$ until growth occurred.

## Statistics and reproducibility

For differential expression analysis, all statistics of this work were performed via the DESeq2 R package, which uses a two-sided Wald test to compare transcriptomics datasets[78]. *p*-values were adjusted using the Benjamini and Hochberg correction. No statistical method was used to predetermine the sample size. No data were excluded from the analyses. The experiments were not randomized. The investigators were not blinded to allocation during experiments and outcome assessment.

## Reporting summary

Further information on research design is available in the Nature Portfolio Reporting Summary linked to this article.

## Data availability

Sequencing raw data (detailed in Supplementary Data 5) have been deposited at the European Nucleotide Archive under project accession code PRJEB79489. The genome sequences of CO-1 and $Ech2_{KI}$ are available under accessions GCA_964276765 and GCA_965140945, respectively. Source data are provided in this paper.

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

## Acknowledgements

The authors are indebted to Marlies Müller, Ivo van den Hurk, Dominic Uhlir, Julia Reichebner and Angeliki Sitara for excellent technical assistance, and Irma Querques for fruitful discussions. This work was supported by the Christian Doppler Research Association and Circe Biotechnologie GmbH, Vienna, Austria. The authors acknowledge TU Wien Bibliothek for financial support through its Open Access Funding Program.

## Author contributions

Conceptualization: R.H. and S.P.; Investigation: R.H., J.H., and M.S.; Formal analysis: R.H., J.H., M.S., M.M., G.G.T., and S.P.; Software: J.H., G.G.T., and M.M.; Visualization: R.H.; Funding acquisition: S.P.; Supervision: S.P.; Writing – original draft: R.H. and M.S.; Writing – review & editing, R.H., J.H., M.S., G.G.T., and S.P.

## Competing interests

During parts of the study, R.H. has been employed by Circe Biotechnologie GmbH, a company with commercial interest in microbial gas fermentation processes. Circe Biotechnologie GmbH has filed a patent application relating to the biocatalyst for CO and syngas fermentation obtained in this study at the Austrian patent office (n° A50333/2023), in which R.H., J.H., and S.P. are listed as inventors. The other authors declare no competing interests.
