## [Peer Review file · Nature Communications]

A megatransposon drives the adaptation of *Thermoanaerobacter kivui* to carbon monoxide

Corresponding Author: Dr Stefan Pflügl

Version 0:

Reviewer comments:

Reviewer #1

(Remarks to the Author)

The authors adapted the acetogen *Thermoanaerobacter kivui* to grow on CO and thoroughly investigated the genetic basis behind this. The capability was previously described, but the genomic reasons for this had not been thoroughly dissected before. This paper describes how the authors discovered that carboxydutrophy is dependent on the mobilization of the megatransposon TnCO-1 ecDNA. This research marks a significant contribution by demonstrating how *T. kivui*, with a rather small genome (~2.4 Mbp), can adapt to grow rapidly on CO. They show that transposases apparently trigger the emergence of an additional circular ~86 kbp sequence in the course of adaptation. This is very exciting, as it showcases that an isolate of an acetogen – a bacterium with a primordial metabolism – which inherently did not have extrachromosomal DNA (ecDNA), can generate ecDNA from its own genome. This is a particularly intriguing finding as the WLP in acetogens does not follow a specific phylogeny, suggesting that it was acquired by different bacterial phyla through horizontal gene transfer. However, extrachromosomal elements that carry the WLP have not been discovered thus far. The observations made in this paper now clearly support the idea that HGT was involved in the spread of the WLP, explaining its wide distribution. Overall, the manuscript describes 2 cases of mobility of the WLP locus: (1) mobilization of Tn and (2) large inversion of the genome, both likely mediated by transposases.

This well-executed research offers new insights into the genomic mechanisms in strict anaerobic microbes that enable them to acquire new metabolic features – in this case carboxydutrophy. This is extremely valuable as these microbes are used for syngas fermentation on an industrial scale (in which optimization is of great value), and at the same time the emergence of new metabolic traits through ecDNA is still largely unexplored. I congratulate the authors for a wonderful piece of work!

Major comments:

1. Can the authors isolate the ~86 kbp circular ecDNA element and transform it into *T. kivui* before adaptation (H2?) < is this strain now able to grow on CO?
2. If ONT data is available, can the authors look at the DNA methylation patterns (using Rerio or Nanodisco?) to see if the mods are the same for the main chromosome and the ecDNA? This is likely the case, as the authors note an increase in TnCO-1-specific amplicons, suggesting a higher ecDNA copy number. How is this regulated? By a different modification motif? This would be extremely interesting to determine, especially since to the reviewer's knowledge there are no high copy number plasmids available for acetogens and this would be great for industrial applications.
3. Is the megatransposon region still present in the main chromosome or completely excised (sorry if I missed this)?

Minor comments:

l. 27 "homoacetogens" is not commonly used anymore. Please exchange with "acetogens"

l. 33 Microbial biomass?

l. 37 not all acetogens are autotrophs

l. 59 onward: this is unclear

l. 96 "The" CO concentration was...

Figure 1 is at quite low resolution and quite complex. Number of transfers would be good to indicate and please revise panel

A so it is easier to understand.

l. 126: particularly high in comparison to what?

Figure 2: Why is it important to show panels C and D? Please explain this in the text. Why are there no error bars in these panels?

l. 150 + 240 + 362: not sure if it ok in this journal to state "data not shown"?

l. 221: was the overall sequencing depth the same? What was it?

l. 306: consider exchanging "born" with "harbored" as this seems confusing and maybe even wrong?

Figure 6: What does the "FC" in the heatmap / left box (next to G-1 and CO-1) mean? Are these mean TPM values? Please explain this (e.g. in figure legend)?

l. 389-391: are there SNVs/InDels in the pleiotropic regulators?

l. 454: Please state here (again) what Ech2 KI refers to, to make it easier to understand for the reader

l. 694: pelleted and shock-frozen

Reviewer #2

(Remarks to the Author)

The paper is well-written, with a clear and thorough presentation of the research. The authors successfully adapted the thermophilic acetogen *Thermoanaerobacter kivui* to use CO as its sole carbon and energy source. They identified a CO-1 isolate that exhibited rapid growth on CO and syngas in both batch and continuous cultures ($\mu_{max} \sim 0.25 \text{ h}^{-1}$). The study revealed that the acquisition of carboxydutrophy is intrinsically linked to the mobilization of a circular, CO-inducible, 86 kb megatransposon originating from the 79 WLP locus. Transcriptomic analysis highlighted the importance of maintaining redox balance during carboxydutrophic growth.

The authors commendably validated their hypothesis through genetic modifications, overcoming the well-known challenges of engineering acetogens. This is a significant achievement, demonstrating a robust approach to address complex biological questions.

While I will not suggest measuring ferredoxin levels due to the difficulty of this task, it would be valuable to explore the role of hydrogenase inhibition by CO, given the observation that growth on H_2/CO_2 and low-CO syngas was possible across all populations and clones.

Like in other acetogens, the role of Ech1 in *T. kivui* metabolism is not yet fully understood, with no Fd_2^- oxidation activity identified for Ech1. However, the observed upregulation of Ech2 genes suggests a potential CO tolerance mechanism, possibly resembling that of CO-1, where balancing the Fd_2^-/Fd pool appears crucial. Further exploration of these mechanisms would provide valuable insights into the metabolic adaptation of *T. kivui* to CO and help advance our understanding of redox management in acetogens.

It would be interesting if the authors could further explore the mechanism by which TnCO-1 mobilizes and rearranges two genomic loci into a single circular DNA molecule. Specifically, clarifying whether this occurs through a copy-in-paste-out mechanism involving circularization and subsequent excision, or through separate mobilization and fusion events, would enhance the understanding of IS1182-like elements. Given the remarkable stability of TnCO-1 mobilization in CO-1 upon CO exposure, further investigation into the triggers and regulation of this process could yield deeper insights into the genetic and metabolic adaptation of *T. kivui* to carboxydutrophy.

Reviewer #3

(Remarks to the Author)

Review:

A megatransposon drives the adaptation of *Thermoanaerobacter kivui* to carbon monoxide (Manuscript ID. NCOMMS-24-60485)

Article summary:

In this article, the authors adapted *Thermoanaerobacter kivui*, a thermophilic acetogen, to use CO as the sole carbon and energy source through adaptive laboratory evolution (ALE). Long-read sequencing of the adapted strain CO-1 and coverage analysis revealed that an extrachromosomal circular megatransposon containing genes involved in the Wood-Ljungdahl pathway (WLP) and energy conservation allowed the strain to acquire carboxydutrophy (capability to grow on CO). With several validation experiments, including transcriptome analysis and reverse engineering, the authors validated the hypothesis that carboxydutrophy was acquired by this megatransposon.

Transcriptome analysis of the CO-1 strain revealed upregulation of Ech2 and downregulation of CODH, both of which are involved in redox balancing, particularly in the regulation of Fd or Fd_2^- generation. Based on this finding, the authors argued that balancing Fd/ Fd_2^- pool is a key to gaining carboxydutrophy in CO-1 strain. To validate the hypothesis, they constructed a rationally engineered strain by overexpressing Ech2 in the wild-type *T. kivui* strain and subsequently adapted the engineered strain under CO conditions. Consistent with the earlier findings, genome rearrangement event was reproducible but in different manner where a large inversion of the genomic region bearing the WLP locus occurred in the Ech2-overexpressing strain.

Overall, the results of this study provide a somewhat novel report, offering new insights into the mobility of the WLP locus and its association with the adaptation mechanism of homoacetogens that acquire the ability to grow on CO. While the authors made efforts to validate their hypothesis through several experiments, there are still some points that need to be addressed to fully elucidate the mechanism behind megatransposon occurrence in *T. kivui*. More detailed explanations and

descriptions of the data are necessary. Addressing these points would further improve the quality of the article.

Comments:

1. One of the major topics of this study is CO metabolism in *T. kivui*. However, there is insufficient background explanation on the metabolic process and energy conservation pathways when CO is used as a substrate. Additionally, more specific and stoichiometric details about the advantages of using CO over other carbon sources could be included in the introduction section.
2. Since the authors emphasize the potential of the adapted strain as a syngas bioprocessor (Discussion section), comparing metabolic profiling data other than acetate between WT and new strains (CO-1, Ech2ki) should be given to provide more convincing results.
3. Extrachromosomal mega-transposon, Tn-CO-1, is described to be crucial for carboxydutrophy. However the article provides insufficient speculations and lacks related references regarding its construction and maintenance mechanism. The background information and biological meaning of them should be included in Result or Discussion section with more details.
4. Does the megatransposon observed in the strain truly originate from the transposase sequence? Validation experiments, such as knocking out the transposase genes (IS1182-like) or their recognition sequences and examining changes in phenotype, could be considered.
5. In Figure 4a. the intensity of TnCo-1 specific amplicons appears to be higher under glucose conditions compared to H₂/CO₂ conditions. A more detailed comparison of amplicon intensities across all growth conditions (CO, H₂/CO₂, and glucose), rather than focusing solely on CO conditions, is necessary to avoid misinterpretation of the data.
6. The underlying mechanisms that explain why the transformation of empty vectors leads to the elimination of megatransposon in the strain should be clarified, as well as a rationale for choosing these methods for elimination (sentence 235-252).
7. In sentence 339-372, the rearrangement of the WLP locus in Ech2-ki strain, positioning it closer to origin of replication was assumed to cause the increase in its expression levels. Conducting RT-qPCR of the specific region or transcriptome analysis of this strain appears necessary to quantitatively compare the expression levels with those in WT and CO-1 strains. This could strengthen the authors' hypothesis about the arrangement, rather than merely inferring it.
8. The CO-inducible megatransposon mobilization is understandable, but a plausible mechanism explaining why megatransposon generation is induced specifically by CO, and not by glucose or H₂/CO₂, should be provided.
9. In Figure 6 and the related paragraph, the transcriptome patterns of the adapted strain and wild-type strain were compared under different conditions: the wild-type grown in H₂/CO₂ and the adapted CO-1 strain grown in CO. Since energy metabolism and redox balancing mechanisms can vary between these gaseous substrates, particularly due to the energy source differences (H₂ or CO), it may be necessary to use the same growth conditions for a proper comparison of the transcriptomes of the two strains. Alternatively, a different experimental design for the transcriptome analysis conditions should be considered.

Reviewer #4

(Remarks to the Author)

The authors carry out a significant body of work to adapt the acetogen *Thermoanaerobacter kivui* to use CO as sole carbon and energy source. Their interdisciplinary approach involves laboratory evolution, batch and chemostat gas fermentation experiments, functional genomics, genetic engineering, and bioinformatics. Importantly, they show that the carboxydutrophic phenotype was facilitated by the mobilization of a CO-inducible megatransposon. Transcriptomics further suggests that potentially the elevated expression of the hydrogenase Ech2 thanks to the megatransposon could be the functional link between the latter and carboxydutrophy of *T. kivui*. The authors thus create an Ech2 knock-in strain and identify major genomic rearrangements that support growth on CO as sole carbon and energy source. However, there are limitations regarding the Ech2 strain results and the overarching theory (see general comments) that would need addressing for drawing a reliable conclusion on the possible functional mechanism how the megatransposon facilitates carboxydutrophy. In other words, it is not clear currently whether Ech2 expression is the key factor. These tasks need to be performed, in our opinion, before disseminating this work to the public. If the theory is supported by the additional results, this work will be a landmark for understanding and engineering acetogen metabolism. It would also be relevant for the general microbiology and metabolic engineering communities. Overall, the work meets the standards in the field but specific comments need to be addressed to sufficiently describe methodology and data analysis.

General comments

1. The results and claims made about the effect of Ech2 knock-in are not convincing currently. Firstly, no experimental data currently show elevated expression of Ech2 in the KI strain. Please provide qPCR or protein expression results to show elevated Ech2 expression in the Ech2KI strain vs the PyrEK1pop culture (WT is not a correct control as Ech2KI is supplied with uracil that can affect growth and Ech2KI went through sub-culturing). It would also be interesting to see Ech2

expression in Ech2KI vs CO-1. Secondly, Ech2KI mutation results in Lines 383-420 are presented vs the WT (non-uracil medium). Growth uracil and sub-culturing of the population and selection during one month prior to clone isolation could introduce mutations not triggered by Ech2 knock-in (as was the case when H-2, H-1 and CO-1 were isolated). Thus to support the claims about potential roles of the mutations, the authors would need to compare Ech2KI mutation data vs PyrEKLpop or isolates that have been sub-cultured/selected on uracil for the same period. Thirdly, please clearly state that the TnCO-1 megatransposon was not detected in the Ech2KI isolate (if that was the case). This would be a strong argument for the megatransposon regulating Ech2 expression (i.e. not needed when Ech2 increased through other means), and would also raise the interesting question of how is megatransposon formation triggered. Lastly, in line 416 the authors suggest stronger expression of WLP is beneficial while RNA-seq data in lines 315-321 shows repression of CODHs – please explain.

2. The authors propose that elevated Ech2 expression in CO-1 is the main cause behind better growth on CO through higher consumption of Fd₂- by the Ech2. This is likely and this critical hypothesis could be verified by an easy fermentation experiment – do an agitation ramp (or also gas flow ramp if mass transfer not limited by agitation) during steady-state growth of CO-1 in chemostat and see if H₂ production increases proportionally with CO uptake. And compare this effect on syngas to WT (where WT can consume CO).

3. The data seems not to support the claim that CO-1 shows co-utilisation of CO, CO₂, and H₂. See below and address.

Specific comments

1. Line 24: Reads a bit off, suggested to change to “of maintenance of redox balance...”

2. Line 25: If word count allows, would recommend to also mention phenotypes beyond ability to grow on CO (u_{max}, max OD etc).

3. Line 38: Would recommend to use a reference(s) to the original papers working on/describing the pathway, e.g. DOI: 10.1096/fasebj.5.2.1900793

4. Line 48: Would recommend to add commas “..., to a much lower extent...”

5. Line 99: Please explain why such a lower number of clones were isolated

6. Figure 1b: Are these max ODs for all? If yes, how was this determined

7. Line 127: Would be relevant to add WT u_{max} on syngas.

8. Figure 2: Add x-axes info for a and b; specify in legend or in methods how was u_{max} calculated; denote in legend if points are average or mean; why are c and d missing error bars?; on c, d, if spikes at end are not biological, remove these datapoints

9. Line 130: Figure2c does not show positive values for CO₂ uptake, so where is co-utilisation for the three gases?

10. Line 150: Why are data for CO-2 and CO-3 not shown? They would strengthen the approach

11. Line 185: Why wasn't PCR ran for WTpop as G-1 could just be one clone from a genetically heterogeneous population?

12. Lines 213 – 214 /Figure 4 – The authors state that abundance of the Tnco-1 specific amplicons increased on CO referring to the semi-quantitative PCR results on Figure 4a. However, based on the gel results, the same could be said about the glucose-grown cultures if compared to the H₂CO₂ samples. As the PCR experiment is followed by a solid sequencing approach which supports the claim of CO-dependent Tnco-1 abundance, please address the prominent Tnco-1 bands from the glucose-grown cultures (which currently undermine the main point) on panel a in the text. For results section focusing on Figure 4b - it would be helpful to see the number of reads/Gb for all discussed comparisons and sequencing timepoints shown on the figure. In addition, please comment on the result showing no Tnco1 expression in CO1-G8 but then showing low expression in CO1-G11.

13. Line 218: “grow back” is a weird construct, please improve

14. Line 222: Quantify “much higher”

15. Line 242 – “..neomycin-resistant strains...” – clones or cells.

16. Line 284: Comment on acetate yield vs D=0.1

17. Line 287: So wash-out was detected above D=0.25?

18. Chemostat runs: Did you try to achieve higher biomass levels?

19. Table 2: Specify what do values before and after ± show

20. Lines 315-317: The link between the two sentences is not clear – first says mono-CODHs are repressed, second says CODH is essential. Please clarify.

21. Line 318: Monofunctional CODHs are not using CO directly?

22. Line 367 – “A single isolate...” – Why was only a single isolate sequenced?

23. Lines 385 – 387 – The hypothesis about potential changes in Ech2 subunit expression levels is understandable, but seems out of place amidst a section on chromosomal rearrangements.

24. Lines 390 – 391 – “...could modulate carbon metabolism.” Could as in have been shown to regulate carbon metabolism? If yes, in which organisms?

25. Line 398 – “...multiple loci...” Please state how many.

26. Figure 7c and 7d – Please discuss the lower max OD of Ech2KI compared to CO-1, as this is important. Especially in light of multitude of genomic rearrangements in the KI strain. Also, the figure could benefit from making the bar charts larger for easier inspection. Please specify in the figure legend that the red-colored part of the chromosome on 7d marks the inversion event.

27. Lines 425 – 426 – How would you explain such a drastic difference in T_d between the CO-1 strain and the previously CO-adapted *T. kivui* by Weghoff & Müller? Please address in the text.

28. Lines 454 – 455 – “...our study demonstrated Ech2 KI could kickstart growth on CO...” A little difficult to claim that, as the KI strain carries so many additional mutations. Please dial down, suggestion to “...demonstrated that Ech2 KI could lead to growth on CO” .

29. Line 492-495: Would be relevant to add DOI: 10.1016/j.cej.2022.137678 as reference and expand sentence

30. Lines 528-545: It can be assumed that all this text applies to *T. kivui*, so best to specify that to avoid the confusion whether they also apply for *E. coli*

31. Line 544: Define shaking speed
32. Line 556: Specify/define "high growth"
33. Line 557: Specify if colonies were grown autotrophically
34. Line 579: State the dilution rates used
35. Lines 583-587: Specify mobile phase, flow rate, column temperature etc.
36. Line 589: Define how steady-state was determined
37. Line 603-...: Specify parameters for pelleting (how much biomass used, centrifugation etc)
38. Line 611: Specify read length
39. Line 633: "was" missing between "sequence" and "manually"
40. Line 656: "was" missing after "chromosome"; "these" instead of "this"
41. Line 686 - Genotyping – Please add more details to the semi-quantitative PCR approach. How much DNA was used, number of cycles, etc. or reference the protocol used in this work.
42. Line 694: "pelleted" instead of "pellets"; specify parameters for pelleting (how much biomass used, centrifugation etc)
43. Line 697: Specify conditions for lysis
44. Line 698: Please note how was RNA integrity evaluated?
45. Line 764: Accession number missing

Reviewer #5

(Remarks to the Author)

Version 1:

Reviewer comments:

Reviewer #1

(Remarks to the Author)

The authors have adequately addressed this reviewer's questions and provided the necessary revisions to strengthen the manuscript. I thus endorse the manuscript for publication in Nature Communications.

Reviewer #2

(Remarks to the Author)

I have reviewed the resubmission of A megatransposon drives the adaptation of Thermoanaerobacter kivui to carbon monoxide. The authors have addressed my concerns as well as those of the other reviewers. I recommend that the manuscript be published.

Reviewer #3

(Remarks to the Author)

The authors have fully addressed the suggested comments by incorporating several detailed descriptions and supplements: discussing CO metabolism of T. kivui in the introduction, its characteristics as a biocatalyst, and providing a detailed explanation of plausible mechanisms for transposition in the discussion. Additionally, they have moderately described TnCO-1 maintenance, clarified the rationale for the TnCO-1 elimination method, and conducted additional transcriptomics experiments and analyses that support the carboxydrotrophic phenotype of T. kivui. The revised manuscript has improved in clarity and is now better supported by additional data and explanations.

Reviewer #4

(Remarks to the Author)

The authors have addressed all but one comment adequately. In addition, please check the reference to Supplementary Figure S3. After addressing the latter, I support publication of this work. I congratulate the authors on this landmark work!

General comments

1. The results and claims made about the effect of Ech2 knock-in are not convincing currently. Firstly, no experimental data currently show elevated expression of Ech2 in the KI strain. Please provide qPCR or protein expression results to show elevated Ech2 expression in the Ech2KI strain vs the PyrEKIpop culture (WT is not a correct control as Ech2KI is supplied with uracil that can affect growth and Ech2KI went through sub-culturing). It would also be interesting to see Ech2 expression in Ech2KI vs CO-1. Secondly, Ech2KI mutation results in Lines 383-420 are presented vs the WT (non-uracil medium). Growth uracil and sub-culturing of the population and selection during one month prior to clone isolation could introduce mutations not triggered by Ech2 knock-in (as was the case when H-2, H-1 and CO-1 were isolated). Thus to support the claims about potential roles of the mutations, the authors would need to compare Ech2KI mutation data vs PyrEKIpop or isolates that have been sub-cultured/selected on uracil for the same period.

Please note that uracil was not used during the selection procedure, as we conversely selected for uracil prototrophy (this

confusion likely stemmed from the “minus” sign used in the figure, that could be interpreted as a “dash” – the figure was edited to avoid this confusion). The PyrEKIpop culture “died” during co-selection on CO + CDM. However, we have an isolated clone PyrEKI from the direct plating selection step. We cultivated that clone, Ech2KI and CO-1 using syngas in serum bottles until the log phase, and performed RNA-seq. The resulting data showed unambiguously that the ech2-related genes were upregulated in Ech2KI (Supplementary Figure S4, Supplementary Table S7) and, even more so, that ech2 overexpression is the main difference arising between Ech2KI and PyrEKI at the transcriptional level. We added these results at lines 473-485.

It seems line 476 mistakenly refers to Supplementary Figure S3, instead of Supplementary Figure S4.

28. Lines 454 – 455 – “...our study demonstrated Ech2 KI could kickstart growth on CO...” A little difficult to claim that, as the KI strain carries so many additional mutations. Please dial down, suggestion to “...demonstrated that Ech2 KI could lead to growth on CO” .

Because selection for carboxydrotrophic growth yielded a strain with a correct integration of the ech2 overexpression cassette (as opposed to selection for uracil prototrophy, Figure 7), we do not believe this statement needs to be dialed down, particularly in the light of the novel results reported in this revised manuscript.

The novel results do not change the fact that the KI strain carries many additional mutations. Also in the new Supplementary Figure S4, there are genes that do not belong to the ech2 locus but expressed equally highly – orange dots among the green ech2 locus “cloud”. So it would be fair to dial down.

In addition, please identify those most up-regulated (orange) genes part of the ech2 cloud on the volcano plot as well as the 5 most down-regulated genes.

Reviewer #5

(Remarks to the Author)

Version 2:

Reviewer comments:

Reviewer #4

(Remarks to the Author)

The authors have addressed my comments and I endorse publication. Congratulations!

Reviewer #5

(Remarks to the Author)

Response to reviewers

We would like to sincerely thank all reviewers for the time invested in thoroughly evaluating our manuscript, and for their feedback on our work.

In the revised version of the manuscript, we believe we addressed the reviewers' concerns, notably by performing new major experimental and computational work that further supports our conclusions. In particular, we have added a set of chemostats and related transcriptomic analyses of strains G-1 and CO-1 grown with low (30 %) CO syngas to prove that the upregulation of *ech2* genes is strain-dependent (Supplementary Figure S3). The fermentation data also undoubtedly show that H₂, CO₂ and CO are co-consumed (updated Table 2). Additionally, we performed a comparative transcriptomic analysis of the strains PyrE_{KI}, CO-1 and Ech2_{KI} (described in Figure 7) in serum bottles grown with syngas (25 % CO). This not only shows that rationally overexpressing the *ech2* complex of *T. kivui* was successful but also further demonstrates that the major difference between carboxydrotrophic (CO-1, Ech2_{KI}) and non-carboxydrotrophic (PyrE_{KI}) strains lies in the upregulation of the *ech2* genes (Supplementary Figure S4, Supplementary Table S7). We also provide quantitative data showing growth on 100 % CO for the strains CO-2 and CO-3 (Supplementary Figure 1), for which Tn_{CO-1} was also detected. Finally, we explored the IS1182-mediated mobilization of Tn_{CO-1} using the ISfinder transposon database and the literature, and offer a potential explanation on the molecular process yielding Tn_{CO-1} in the CO-1 strain (Supplementary Figure 5).

Taken together, these new results strengthen our previous observations and further emphasize the crucial importance of the mobilization of Tn_{CO-1} and of the upregulation of *ech2* for carboxydrotrophic growth in *T. kivui*.

REVIEWER COMMENTS

Reviewer #1 (Remarks to the Author):

The authors adapted the acetogen *Thermoanaerobacter kivui* to grow on CO and thoroughly investigated the genetic basis behind this. The capability was previously described, but the genomic reasons for this had not been thoroughly dissected before. This paper describes how the authors discovered that carboxydrotrophy is dependent on the mobilization of the megatransposon TnCO-1 ecDNA. This research marks a significant contribution by demonstrating how *T. kivui*, with a rather small genome (~2.4 Mbp), can adapt to grow rapidly on CO. They show that transposases apparently trigger the emergence of an additional circular ~86 kbp sequence in the course of adaptation. This is very exciting, as it showcases that an isolate of an acetogen – a bacterium with a primordial metabolism – which inherently did not have extrachromosomal DNA (ecDNA), can generate ecDNA from its own genome. This is a particularly intriguing finding as the WLP in acetogens does not follow a specific phylogeny, suggesting that it was acquired by different bacterial phyla through horizontal gene transfer. However, extrachromosomal elements that carry the WLP have not been discovered thus far. The observations made in this paper now clearly support the idea that HGT was involved in the spread of the WLP, explaining its wide distribution. Overall, the manuscript describes 2 cases of mobility of the WLP locus: (1) mobilization of Tn and (2) large inversion of the genome, both likely mediated by transposases.

This well-executed research offers new insights into the genomic mechanisms in strict anaerobic microbes that enable them to acquire new metabolic features – in this case carboxydrotrophy. This is extremely valuable as these microbes are used for syngas fermentation on an industrial scale (in which optimization is of great value), and at the same time the emergence of new metabolic traits through ecDNA is still largely unexplored. I congratulate the authors for a wonderful piece of work!

We would like to warmly thank reviewer 1 for the very appreciative comments!

Major comments:

1. Can the authors isolate the ~86 kbp circular ecDNA element and transform it into *T. kivui* before adaptation (H2?) < is this strain now able to grow on CO?

We originally tried this approach, which did not work. The main reason is likely that we have not been able to purify DNA with a sufficiently high MW (e.g., N50 of 5-15 kb, enough for ONT *de novo* sequencing, but far too low for the transposon size).

2. If ONT data is available, can the authors look at the DNA methylation patterns (using Rerio or Nanodisco?) to see if the mods are the same for the main chromosome and the ecDNA? This is likely the case, as the authors note an increase in TnCO-1-specific amplicons, suggesting a higher ecDNA copy number. How is this regulated? By a different modification motif? This would be extremely interesting to determine, especially since to the reviewer's knowledge there are no high copy number plasmids available for acetogens and this would be great for industrial applications.

We basecalled methylation modifications from ONT sequencing using dorado with the latest high accuracy models to accommodate our R10.1 raw data. modkit was used to evaluate methylation percentage for the following modifications: 6mA, 5mCpG, 5hmCpG, 5mC, 5hmC, and 4mC. We compared methylation patterns of the whole chromosome (WT), the megatransposon (TnCO-1) and the region of the chromosome homologous to TnCO-1 (HR) region. As the sequence of TnCO-1 and HR are almost identical and only reads that uniquely map to either HR or TnCO-1 can be used to assess methylation, major parts of the regions remain uncovered (Figure 1).

Figure 1. Sequencing coverage profile of TnCO-1 and HR based on uniquely mapped reads. Maximum coverage is ~100-fold.

Summarizing methylation information of the WT, TnCO-1 and HR revealed that adenosine methylation was the most prevalent modification as nearly six percent of adenines found in the genome were methylated (Table 1). The second most common methylation type was 4mC: around two percent of cytosines were methylated. Furthermore, regardless of context, cytosine 5-methylation was found to be rare.

Table 1. Summary of the different modification levels between WT, TnCO-1, and HR based on modification type (MOD), the number of all available nucleotides for modification type (N_NUC), the number of methylated loci (≥ 1 bases methylated) pre-/post-filtering for (N_ML, N_F_ML, filter=COV ≥ 3) number of modified bases (N_MOD) filtered base coverage (COV), percentage of methylated loci lost due to filtering (%LOSS_ML), percentage of modified bases (%MOD), percentage of coverage lost due to filtering (%LOSS_C).

CHR	MOD	N_NUC	N_ML	N_F_ML	N_MOD	%LOSS_M	%MOD
WT	5m-CpG	132 682	5434	5427	7 305	0,13%	0,22%
TnCO-1	5m-CpG	4 870	30	27	28	10,00%	0,22%
HR	5m-CpG	4 864	34	31	46	8,82%	0,21%
WT	5hm-CpG	132 682	6845	6834	9 096	0,16%	0,28%
TnCO-1	5hm-CpG	4 870	33	30	33	9,09%	0,26%
HR	5hm-CpG	4 864	41	40	47	2,44%	0,22%
WT	6mA	765 468	1121373	1120727	3 587 290	0,06%	5,59%
TnCO-1	6mA	26 036	9365	8225	14 123	12,17%	5,54%
HR	6mA	26 080	11423	10779	21 780	5,64%	5,25%
WT	5mC	441 326	129419	129311	177 602	0,08%	0,48%
TnCO-1	5mC	18 505	653	600	697	8,12%	0,43%
HR	5mC	18 487	945	891	1 117	5,71%	0,44%
WT	5hmC	441 326	79619	79509	100 062	0,14%	0,27%
TnCO-1	5hmC	18 505	382	351	402	8,12%	0,25%
HR	5hmC	18 487	547	529	639	3,29%	0,25%
WT	4mC	441 326	355771	354602	640 349	0,33%	1,96%
TnCO-1	4mC	18 505	2263	2008	2 615	11,27%	1,85%
HR	4mC	18 487	3028	2860	4 128	5,55%	1,83%

Looking specifically at a sequence range, which is covered in both the HR and TnCO-1, we did not see any significant difference in methylation patterns (Figure 2).

Figure 2. Methylation patterns of HR (1,919,109..1,924,109) and corresponding TnCO-1 (70,807..75,807) with uniquely mapped reads. (A) CpG methylation; (B) Adenosine methylation.

Based on this, it seems unlikely that methylation is involved in formation of TnCO-1 or regulation of its copy number. Based on the coverage (lines 188-189), TnCO-1 is not present at high copy in the CO-1 strain.

3. Is the megatransposon region still present in the main chromosome or completely excised (sorry if I missed this)?

Based on our sequencing data, the megatransposon sequence is present both in the main chromosome and as an ecDNA. This was further emphasized at lines 201-203.

Minor

comments:

I. 27 “homoacetogens” is not commonly used anymore. Please exchange with “acetogens”

The term was replaced.

I. 33 Microbial biomass?

Biomass refers to plant biomass here; this was clarified in the text.

I. 37 not all acetogens are autotrophs

Autotroph was removed.

I. 59 onward: this is unclear

We reformulated that paragraph for clarity (now lines 72-86).

I. 96 “The” CO concentration was...

Figure 1 is at quite low resolution and quite complex. Number of transfers would be good to indicate and please revise panel A so it is easier to understand.

The text and figure were edited. High resolution figures (1000 dpi) were included (in the merged file, see figures at the end).

I. 126: particularly high in comparison to what?

“Particularly high” was removed, as the comparison to the literature is covered in the discussion part.

Figure 2: Why is it important to show panels C and D? Please explain this in the text. Why are there no error bars in these panels?

We think it is important to show a consumption profile of the feedstocks used for the continuously sparged batch cultivations. In case of syngas, it demonstrates that CO and H₂ are co-consumed, while CO₂ is produced and that CO fermentation displays a water gas shift reaction with net H₂ production.

We have revised the text with an explanation. The figure now shows error bars of gas consumption/production rates.

I. 150 + 240 + 362: not sure if it ok in this journal to state “data not shown”?

line 150: we added a new Supplementary Figure (S1).

Line 240: we added a reference to a new preprint, in which we optimized the transformation protocol of *T. kivui*.

Line 362: these negative results are indeed not included in the current version of the manuscript. However, please find below an agarose gel showing amplifications on gDNA at the *pyrE* locus for 7 clones randomly selected from the *ech2* KI transformation plate (direct plating without co-selection with CO, as shown in figure 7c, top plate). This gel shows no integration for 3 clones (lanes 2, 3, 7), partial integration for 3 clones (4, 5, 8), and one unsuccessful amplification (lane 6). $\Delta pyrE$ control shows an unspecific amplification at 9 kb, presumably because of the extra-long extension time of the PCR (9 min) compared to the size of the desired amplicon for the control (2.1 kb). The PCR was later optimized to remove the unspecific product (see Fig. 7C, bottom gel). Strains corresponding to lanes 4, 5, 6 and 8 were tested for carboxydrotrophy, with PyrEKI as control. None of them could grow with CO as sole carbon/energy source (15 days, 50 % CO).

Figure 3. Genotyping of clones obtained directly after transformation of the Ech2_{KI} DNA template. PCR on gDNA targeting the *pyrE* locus, that is, the insertion target. Controls: PyrEKI (KI of *pyrE* without *ech2* subunit genes), G-1 (WT), G-1 Δ *pyrE* (recipient strain), NTC: no template control. Expected sizes are indicated with an arrow.

I. 221: was the overall sequencing depth the same? What was it?

As shown in Supplementary Table S6, the sequencing depth ranged from 74X to 444X. This is why we normalized using the sequencing depth (values are shown as Reads per Gb). For clarity, this was added in the results (lines 251-252), in addition to the methods section (already in original manuscript).

I. 306: consider exchanging “born” with “harbored” as this seems confusing and maybe even wrong?

There was a typo here, we meant borne and not born (that is, in the sense of carry). This was corrected in the text.

Figure 6: What does the “FC” in the heatmap / left box (next to G-1 and CO-1) mean? Are these mean TPM values? Please explain this (e.g. in figure legend)?

FC_{CO} correspond to the fold change calculated by DESeq2 within the differential expression analysis (comparing CO-1 and G-1 replicates). This was added in the legend. All values from figure 6 are taken from Supplementary table S3 (also added in the legend).

I. 389-391: are there SNVs/InDels in the pleiotropic regulators?

As written in the text, *codY* and *phoU* were mutated in Ech2_{KI}. The cognate mutations were added in the text for clarity.

I. 454: Please state here (again) what Ech2 KI refers to, to make it easier to understand for the reader

We clarified the corresponding sentence in the text.

I. 694: pelleted and shock-frozen

We edited the corresponding words.

Reviewer #2 (Remarks to the Author):

The paper is well-written, with a clear and thorough presentation of the research. The authors successfully adapted the thermophilic acetogen *Thermoanaerobacter kivui* to use CO as its sole carbon and energy source. They identified a CO-1 isolate that exhibited rapid growth on CO and syngas in both batch and continuous cultures ($\mu_{\max} \sim 0.25 \text{ h}^{-1}$). The study revealed that the acquisition of carboxydrotrophy is intrinsically linked to the mobilization of a circular, CO-inducible, 86 kb megatransposon originating from the 79 WLP locus. Transcriptomic analysis highlighted the importance of maintaining redox balance during carboxydrotrophic growth.

The authors commendably validated their hypothesis through genetic modifications, overcoming the well-known challenges of engineering acetogens. This is a significant achievement, demonstrating a robust approach to address complex biological questions.

While I will not suggest measuring ferredoxin levels due to the difficulty of this task, it would be valuable to explore the role of hydrogenase inhibition by CO, given the observation that growth on H_2/CO_2 and low-CO syngas was possible across all populations and clones.

We would like to thank reviewer 2 for the positive evaluation of our manuscript. We also agree exploring the role of CO in hydrogenase inhibition would be valuable to get more quantitative insights into the toxicity of CO for this particular microorganism. However, it is clear from literature that CO strongly inhibits Fe-Fe hydrogenases (see for instance scheme 8 from Tard and Pickett, Chem. Rev. 2009, <https://doi.org/10.1021/cr800542q>, or the characterization of the *A. woodii* HDCR from the Müller group <https://doi.org/10.1128/AEM.01772-15>), which is likely the cause for CO toxicity in *T. kivui*.

Like in other acetogens, the role of Ech1 in *T. kivui* metabolism is not yet fully understood, with no Fd_2^- oxidation activity identified for Ech1. However, the observed upregulation of Ech2 genes suggests a potential CO tolerance mechanism, possibly resembling that of CO-1, where balancing the Fd_2^-/Fd pool appears crucial. Further exploration of these mechanisms would provide valuable insights into the metabolic adaptation of *T. kivui* to CO and help advance our understanding of redox management in acetogens.

We entirely agree with the reviewer's comment.

It would be interesting if the authors could further explore the mechanism by which TnCO-1 mobilizes and rearranges two genomic loci into a single circular DNA molecule. Specifically, clarifying whether this occurs through a copy-in-paste-out mechanism involving circularization and subsequent excision, or through separate mobilization and fusion events, would enhance the understanding of IS1182-like elements.

TnCO-1 bears only one transposase gene, i.e., the IS1182-like element. This was confirmed by Blastn in the online tool ISfinder (<https://www-is.biotoul.fr/index.php>). Therefore, as mentioned in the discussion, separate mobilization and fusion of both DNA fragments is unlikely. We further reviewed the scarce literature dedicated to IS1182-like elements. The related transposases have relatively heterogeneous properties. They can drive transposition events with or without passenger genes, with and without duplication of the target insertion site, with or without deletion at the target insertion site, with recognition of a target sequence or structure.

A major common point, however, is that transposition depends on inverted repeat sequences (IR) as recognition sequences for mobilization. Using ISfinder, we found a close IS1182 relative annotated in *Caldanaerobacter subterraneus* subsp. *tengcongensis*. The imperfect IRL and IRR (left and right) matched very well with sequences found directly upstream and downstream *T. kivui* IS1182-like transposase CDS.

An interesting paper from Furi et al. (Front. Microbiol. 2016, <http://doi.org/10.3389/fmicb.2016.01008>) showed the mobilization of a passenger gene linked with the presence of a degenerate IR sequence by an IS1182-like element. When we examined the ends of each duplicated chromosomal region, we found one such sequence at the end of the second duplication (referred to as IRR'). IS1182-mediated transposition using IRL and IRR' as recognition motifs would typically result in a 94 kb circular intermediate if transposition follows a classical copy-out-paste-in process. In this scenario, an additional excision step (of ~9 kb) would yield Tn_{CO-1}.

We could however not find such IR sequences at the P3/P4 junction, and how exactly the corresponding region is excised in Tn_{CO-1} remains elusive. This analysis was included in the discussion (lines 507-530), and represented in the Supplementary Figure S5.

Given the remarkable stability of TnCO-1 mobilization in CO-1 upon CO exposure, further investigation into the triggers and regulation of this process could yield deeper insights into the genetic and metabolic adaptation of *T. kivui* to carboxydrotrophy.

We agree with reviewer 2 that what regulates the mobilization of CO-1 would be highly interesting to know, but we believe that a more detailed molecular investigation of the mobilization process of Tn_{CO-1} is out of the scope of the current manuscript and should be addressed in a future study.

Reviewer #3 (Remarks to the Author):

Review:

A megatransposon drives the adaptation of *Thermoanaerobacter kivui* to carbon monoxide (Manuscript ID. NCOMMS-24-60485)

Article summary:

In this article, the authors adapted *Thermoanaerobacter kivui*, a thermophilic acetogen, to use CO as the sole carbon and energy source through adaptive laboratory evolution (ALE). Long-read sequencing of the adapted strain CO-1 and coverage analysis revealed that an extrachromosomal circular megatransposon containing genes involved in the Wood-Ljungdahl pathway (WLP) and energy conservation allowed the strain to acquire carboxydutrophy (capability to grow on CO). With several validation experiments, including transcriptome analysis and reverse engineering, the authors validated the hypothesis that carboxydutrophy was acquired by this megatransposon. Transcriptome analysis of the CO-1 strain revealed upregulation of Ech2 and downregulation of CODH, both of which are involved in redox balancing, particularly in the regulation of Fd or Fd2-generation. Based on this finding, the authors argued that balancing Fd/Fd2- pool is a key to gaining carboxydutrophy in CO-1 strain. To validate the hypothesis, they constructed a rationally engineered strain by overexpressing Ech2 in the wild-type *T. kivui* strain and subsequently adapted the engineered strain under CO conditions. Consistent with the earlier findings, genome rearrangement event was reproducible but in different manner where a large inversion of the genomic region bearing the WLP locus occurred in the Ech2-overexpressing strain. Overall, the results of this study provide a somewhat novel report, offering new insights into the mobility of the WLP locus and its association with the adaptation mechanism of homoacetogens that acquire the ability to grow on CO. While the authors made efforts to validate their hypothesis through several experiments, there are still some points that need to be addressed to fully elucidate the mechanism behind megatransposon occurrence in *T. kivui*. More detailed explanations and descriptions of the data are necessary. Addressing these points would further improve the quality of the article.

Comments:

1. One of the major topics of this study is CO metabolism in *T. kivui*. However, there is insufficient background explanation on the metabolic process and energy conservation pathways when CO is used as a substrate. Additionally, more specific and stoichiometric details about the advantages of using CO over other carbon sources could be included in the introduction section.

Another paragraph was added in the introduction (lines 58-71).

2. Since the authors emphasize the potential of the adapted strain as a syngas bioprocessor (Discussion section), comparing metabolic profiling data other than acetate between WT and new strains (CO-1, Ech2ki) should be given to provide more convincing results.

Additional information with respect to the potential of *T. kivui* for syngas bioprocessing are given in the respective results section describing the batch and chemostat cultivations. In detail, we provide data showing co-consumption of H₂ and CO in batch cultivations, and co-consumption of H₂, CO and CO₂ in continuous cultures. Moreover, we provide physiological data such as specific gas uptake rate, emphasizing the high catalytic activity of *T. kivui* for bioprocessing of syngas. Combined with the high specific growth rates, this would allow for operation of bioprocesses at high, industrially relevant space-time-yields/productivities. We have emphasized this in the discussion section.

3. Extrachromosomal mega-transposon, Tn-CO-1, is described to be crucial for carboxydotrophy. However the article provides insufficient speculations and lacks related references regarding its construction and maintenance mechanism. The background information and biological meaning of them should be included in Result or Discussion section with more details.

See answer as for reviewer 2 above for the speculations:

Tn_{CO-1} bears only one transposase gene, i.e., the IS1182-like element. This was confirmed by Blastn in the online tool ISfinder (<https://www-is.biotoul.fr/index.php>). Therefore, as mentioned in the discussion, separate mobilization and fusion of both DNA fragments is unlikely. We further reviewed the scarce literature dedicated to IS1182-like elements. The related transposases have relatively heterogenous properties. They can drive transposition events with or without passenger genes, with and without duplication of the target insertion site, with or without deletion at the target insertion site, with recognition of a target sequence or structure.

A major common point however is that transposition depends on inverted repeat sequences (IR) as recognition sequences for mobilization. Using ISfinder, we found a close IS1182 relative annotated in *Caldanaerobacter subterraneus* subsp. *tengcongensis*. The imperfect IRL and IRR (left and right) matched very well with sequences found directly upstream and downstream *T. kivui* IS1182-like transposase CDS.

An interesting paper from Furi et al. (<http://doi.org/10.3389/fmicb.2016.01008>) showed the mobilization of a passenger gene linked with the presence of a degenerate IR sequence by an IS1182-like element. When we examined the ends of each duplicated chromosomal region, we found one such sequence at the end of the second duplication (referred to as IRR'). IS1182-mediated transposition using IRL and IRR' as recognition motifs would typically result in a 94 kb circular intermediate if transposition follows a classical copy-out-paste-in process.

We could however not find such IR sequences at the P3/P4 junction, and how exactly the corresponding region is excised in Tn_{CO-1} remains elusive. This analysis was included in the discussion (lines 507-530), and represented in the Supplementary Figure S5.

4. Does the megatransposon observed in the strain truly originate from the transposase sequence? Validation experiments, such as knocking out the transposase genes (IS1182-like) or their recognition sequences and examining changes in phenotype, could be considered.

There is only one transposase gene encoded in Tn_{CO-1}. As a result, it is highly likely the transposase driving mobilization of Tn_{CO-1} (see previous answer).

The initial reason we introduced plasmid SPF-B017 into CO-1 was as a control to a Tn_{CO-1} and a IS1182 deletion experiment using *T. kivui* endogenous CRISPR system (see Sitara et al. preprint for details). However, it turned out that introduction of the empty backbone was sufficient to abolish carboxydotrophy. As a result of this observation, we do not believe the experiment described by Reviewer 3 can be carried out.

5. In Figure 4a. the intensity of TnCo-1 specific amplicons appears to be higher under glucose conditions compared to H₂/CO₂ conditions. A more detailed comparison of amplicon intensities across all growth conditions (CO, H₂/CO₂, and glucose), rather than focusing solely on CO conditions, is necessary to avoid misinterpretation of the data.

We believe Tn_{CO-1} could behave like a low-copy plasmid, that is, its maintenance/mobilization is selected over time given the appropriate selective pressure (here: CO). In that scenario, it is not surprising that amplicons can still be seen in Figure 4a, while in Figure 4b, longer-term cultivation on glucose shows little to no copies of Tn_{CO-1}. In the case of the different intensities between glucose

and H₂/CO₂, it is possible that the copy number of Tn_{CO-1} decreases faster on H₂/CO₂ than on glucose. This would for example be expected if the maintenance of Tn_{CO-1} has a higher burden on H₂/CO₂ than on glucose. This was discussed in the text (lines 241-245).

6. The underlying mechanisms that explain why the transformation of empty vectors leads to the elimination of megatransposon in the strain should be clarified, as well as a rationale for choosing these methods for elimination (sentence 235-252).

We initially wanted to use a CRISPR tool recently developed by our lab (see Sitara et al., preprint for details) to delete the transposon in the CO-1 strain. This appeared to be unnecessary as introduction of the empty backbone led to the loss of Tn_{CO-1}. We believe that, for an unknown reason, transformation and Tn_{CO-1} are incompatible. Competency might be for example negatively affected by Tn_{CO-1} mobilization, and successful transformation therefore presumably selected a clone lacking Tn_{CO-1}. This was discussed at lines 268-288.

7. In sentence 339-372, the rearrangement of the WLP locus in Ech2-ki strain, positioning it closer to origin of replication was assumed to cause the increase in its expression levels. Conducting RT-qPCR of the specific region or transcriptome analysis of this strain appears necessary to quantitatively compare the expression levels with those in WT and CO-1 strains. This could strengthen the authors' hypothesis about the arrangement, rather than merely inferring it.

We performed transcriptomics and showed that the WLP-genes are not differentially expressed in the Ech2_{ki} strain (Supplementary Figure S4, Supplementary Table S7). The corresponding speculation was emended in the results section (lines 473-485).

8. The CO-inducible megatransposon mobilization is understandable, but a plausible mechanism explaining why megatransposon generation is induced specifically by CO, and not by glucose or H₂/CO₂, should be provided.

We do not believe the megatransposon is induced in the classical sense, in particular because the IS1182-related transposase gene is not differentially expressed in CO-1 (log₂(FC) = 0.8 in the CO-1 (CO) vs G-1 (H₂/CO₂) comparison). Rather, its mobilization could follow a stochastic pattern (i.e., not all cells undergo mobilization), and addition of CO selects for cells mobilizing Tn_{CO-1}. This is however hypothetical, and the possibility of either CO inducibility or CO selection is now stated lines 255-256.

9. In Figure 6 and the related paragraph, the transcriptome patterns of the adapted strain and wild-type strain were compared under different conditions: the wild-type grown in H₂/CO₂ and the adapted CO-1 strain grown in CO. Since energy metabolism and redox balancing mechanisms can vary between these gaseous substrates, particularly due to the energy source differences (H₂ or CO), it may be necessary to use the same growth conditions for a proper comparison of the transcriptomes of the two strains. Alternatively, a different experimental design for the transcriptome analysis conditions should be considered.

We used that particular set of condition as we were interested in what could drive carboxydrotrophy in the new strain. Therefore, a comparison of carboxydrotrophic vs non-carboxydrotrophic conditions made sense, in particular as G-1 is unable to grow solely on CO. However, we agree that these conditions may skew the results towards gas-dependent rather than strain-dependent regulation. We therefore performed additional chemostats under syngas for both the CO-1 and G-1 strain (Table 2). Transcriptomics analysis essentially shows the same behavior, that is, downregulation of the monofunctional CODH and upregulation of Ech2 and Nfn (Supplementary Table 3, Supplementary Figure S3). This was discussed in lines 350-353.

Reviewer #4 (Remarks to the Author):

The authors carry out a significant body of work to adapt the acetogen *Thermoanaerobacter kivui* to use CO as sole carbon and energy source. Their interdisciplinary approach involves laboratory evolution, batch and chemostat gas fermentation experiments, functional genomics, genetic engineering, and bioinformatics. Importantly, they show that the carboxydrotrophic phenotype was facilitated by the mobilization of a CO-inducible megatransposon. Transcriptomics further suggests that potentially the elevated expression of the hydrogenase Ech2 thanks to the megatransposon could be the functional link between the latter and carboxydrotrophy of *T. kivui*. The authors thus create an Ech2 knock-in strain and identify major genomic rearrangements that support growth on CO as sole carbon and energy source. However, there are limitations regarding the Ech2 strain results and the overarching theory (see general comments) that would need addressing for drawing a reliable conclusion on the possible functional mechanism how the megatransposon facilitates carboxydrotrophy. In other words, it is not clear currently whether Ech2 expression is the key factor. These tasks need to be performed, in our opinion, before disseminating this work to the public. If the theory is supported by the additional results, this work will be a landmark for understanding and engineering acetogen metabolism. It would also be relevant for the general microbiology and metabolic engineering communities. Overall, the work meets the standards in the field but specific comments need to be addressed to sufficiently describe methodology and data analysis.

General comments

1. The results and claims made about the effect of Ech2 knock-in are not convincing currently. Firstly, no experimental data currently show elevated expression of Ech2 in the KI strain. Please provide qPCR or protein expression results to show elevated Ech2 expression in the Ech2KI strain vs the PyrEKIpop culture (WT is not a correct control as Ech2KI is supplied with uracil that can affect growth and Ech2KI went through sub-culturing). It would also be interesting to see Ech2 expression in Ech2KI vs CO-1. Secondly, Ech2KI mutation results in Lines 383-420 are presented vs the WT (non-uracil medium). Growth uracil and sub-culturing of the population and selection during one month prior to clone isolation could introduce mutations not triggered by Ech2 knock-in (as was the case when H-2, H-1 and CO-1 were isolated). Thus to support the claims about potential roles of the mutations, the authors would need to compare Ech2KI mutation data vs PyrEKIpop or isolates that have been sub-cultured/selected on uracil for the same period.

Please note that uracil was not used during the selection procedure, as we conversely selected for uracil prototrophy (this confusion likely stemmed from the “minus” sign used in the figure, that could be interpreted as a “dash” – the figure was edited to avoid this confusion). The PyrE_{KIpop} culture “died” during co-selection on CO + CDM. However, we have an isolated clone PyrE_{KI} from the direct plating selection step. We cultivated that clone, Ech2_{KI} and CO-1 using syngas in serum bottles until the log phase, and performed RNA-seq. The resulting data showed unambiguously that the *ech2*-related genes were upregulated in Ech2_{KI} (Supplementary Figure S4, Supplementary Table S7) and, even more so, that *ech2* overexpression is the main difference arising between Ech2_{KI} and PyrE_{KI} at the transcriptional level. We added these results at lines 473-485.

Thirdly, please clearly state that the TnCO-1 megatransposon was not detected in the Ech2KI isolate (if that was the case).

A statement was added lines 453-454.

This would be a strong argument for the megatransposon regulating Ech2 expression (i.e. not needed when Ech2 increased through other means), and would also raise the interesting question of how is megatransposon formation triggered.

As mentioned for Reviewer 3, we believe the megatransposon formation to be generated in a stochastic manner, and CO acts as a selective pressure that eliminates cells that do not mobilize $T_{n_{CO-1}}$.

Lastly, in line 416 the authors suggest stronger expression of WLP is beneficial while RNA-seq data in lines 315-321 shows repression of CODHs – please explain.

The new transcriptomic data in Ech2_{K1} showed the WLP was not differentially expressed and the corresponding statement was emended (line 473-485). As for the monofunctional CODH, we believe its downregulation alleviates the Fd²⁻ overreduction problem (lines 545-546).

2. The authors propose that elevated Ech2 expression in CO-1 is the main cause behind better growth on CO through higher consumption of Fd²⁻ by the Ech2. This is likely and this critical hypothesis could be verified by an easy fermentation experiment – do an agitation ramp (or also gas flow ramp if mass transfer not limited by agitation) during steady-state growth of CO-1 in chemostat and see if H₂ production increases proportionally with CO uptake. And compare this effect on syngas to WT (where WT can consume CO).

To clarify, H₂ production is observed from CO fermentation but not during syngas fermentation. The proposed experiment ramping up the stirrer speed has been performed during generating the data for CO-1 grown on $D = 0.1 \text{ h}^{-1}$ and 0.2 h^{-1} . After increasing the dilution rate to 0.2, we ramped up the stirrer speed to readjust biomass and acetate concentrations to the levels observed at $D = 0.1 \text{ h}^{-1}$ to ensure comparability. While doing that, we did not observe a change in the production of H₂ in relation to CO uptake, i.e. the yield $q_{H_2/CO}$ remained constant. Same is true for growth at different growth rates (0.1 and 0.2 h^{-1}) where q_{CO} increased but the ratio of H₂/CO remained constant (0.208 vs $0.214 \text{ mol H}_2 \text{ mol}^{-1} \text{ CO}$, data from Table 2).

3. The data seems not to support the claim that CO-1 shows co-utilisation of CO, CO₂, and H₂. See below and address.

Thank you for the comment. Co-consumption was indeed not observed in the batch data shown in Figure 2, merely co-consumption of CO and H₂. However, the newly added chemostat data added to Table 2 clearly show co-consumption of CO, H₂ and CO₂ when low CO syngas is used as the gas feed. We revised the description of the batch and chemostat cultivation accordingly.

Specific comments

1. Line 24: Reads a bit off, suggested to change to “of maintenance of redox balance...”

We simplified to “the redox balance”.

2. Line 25: If word count allows, would recommend to also mention phenotypes beyond ability to grow on CO (u_{max} , max OD etc).

The word count unfortunately does not allow for additional words.

3. Line 38: Would recommend to use a reference(s) to the original papers working on/describing the pathway, e.g. DOI: 10.1096/fasebj.5.2.1900793

The reference was added.

4. Line 48: Would recommend to add commas “..., to a much lower extent,...”

Commas were added.

5. Line 99: Please explain why such a lower number of clones were isolated

We directly grew clones in the relevant autotrophic conditions (e.g., high CO syngas or 100% CO). H-1/2, CO-1/2/3 were the ones that grew first.

6. Figure 1b: Are these max ODs for all? If yes, how was this determined

No these are OD measured after 3 days of cultivation, as described in the figure. Starting OD was added in the legend as an additional information.

7. Line 127: Would be relevant to add WT u_{max} on syngas.

The WT does not grow on high (52%) CO syngas, only on low (30%) CO syngas. This was added in the text (lines 144-145).

8. Figure 2: Add x-axes info for a and b; specify in legend or in methods how was u_{max} calculated; denote in legend if points are average or mean; why are c and d missing error bars?; on c, d, if spikes at end are not biological, remove these datapoints

Calculation of μ_{max} was specified in the figure legend, also that points represent the average. Panels c and d: spikes were indeed not biological and have been removed. Error bars have been included.

9. Line 130: Figure2c does not show positive values for CO₂ uptake, so where is co-utilisation for the three gases?

See our response above, major point 3.

10. Line 150: Why are data for CO-2 and CO-3 not shown? They would strengthen the approach

Supplementary Figure S1 was added, in which we show a duplicate serum bottle experiment with CO-1, CO-2 and CO-3. Starting at $OD_{660} = 0.01$ with 2 bar 100 % CO, we grew the strain until biomass started to decrease. The maximal values that were measured are shown in the figure. CO-2 and CO-3 indeed grow on CO as sole carbon and energy source, although they are slower than CO-1. This observation was added in the text (lines 170-171).

11. Line 185: Why wasn't PCR ran for WT_{pop} as G-1 could just be one clone from a genetically heterogeneous population?

G-1 is indeed a single clone from a heterogeneous population. See the newly accepted paper on that matter (Hocq et al. MRA 2025, <https://doi.org/10.1128/mra.01250-24>). Illumina reads of the WT_{pop} were parsed for junction sequences, which confirmed WT_{pop} does not bear Tn_{CO-1}. This was added at lines 215-217.

12. Lines 213 – 214 /Figure 4 – The authors state that abundance of the Tnco-1 specific amplicons increased on CO referring to the semi-quantitative PCR results on Figure 4a. However, based on the gel results, the same could be said about the glucose-grown cultures if compared to the H₂CO₂ samples. As the PCR experiment is followed by a solid sequencing approach which supports the claim of CO-dependent Tnco-1 abundance, please address the prominent Tnco-1 bands from the glucose-grown cultures (which currently undermine the main point) on panel a in the text.

See answer for reviewer 3, reported below:

We believe Tn_{CO-1} could behave like a low-copy plasmid, that is, its maintenance/mobilization is selected over time given the appropriate selective pressure (here: CO). In that scenario, it is not surprising that amplicons can still be seen in Figure 4a, while in Figure 4b, longer-term cultivation on glucose shows little to no copies of Tn_{CO-1} . In the case of the different intensities between glucose and H_2/CO_2 , it is possible the copy number of Tn_{CO-1} decreases faster on H_2/CO_2 than on glucose. This would for example be expected if the maintenance of Tn_{CO-1} has a higher burden on H_2CO_2 than on glucose. This was discussed in the text (lines 241-245).

For results section focusing on Figure 4b - it would be helpful to see the number of reads/Gb for all discussed comparisons and sequencing timepoints shown on the figure. In addition, please comment on the result showing no $Tnco1$ expression in CO1-G8 but then showing low expression in CO1-G11.

The number of reads/Gb will be added in the source data file, as per Nature Communications guidelines.

If Tn_{CO-1} mobilization is random rather than induced (as suggested above), then a low level of reads on glucose is entirely possible.

13. Line 218: “grow back” is a weird construct, please improve

The term was switched to “grow again” (line 249).

14. Line 222: Quantify “much higher”

This was quantified (lines 252-253).

15. Line 242 – “..neomycin-resistant strains...” – clones or cells.

The term was modified to “cells” (line 273).

16. Line 284: Comment on acetate yield vs $D=0.1$

An observation was added (line 325).

17. Line 287: So wash-out was detected above $D=0.25$?

Yes, wash-out was observed once D was increased above 0.25 h^{-1} (statement added lines 327-328).

18. Chemostat runs: Did you try to achieve higher biomass levels?

Not with the conditions shown here.

19. Table 2: Specify what do values before and after \pm show

The values are average \pm standard deviation (statement added in the legend).

20. Lines 315-317: The link between the two sentences is not clear – first says mono-CODHs are repressed, second says CODH is essential. Please clarify.

This is why it is counter-intuitive, as stated in the text. This was emphasized line 363. The underlying theory is that CODH is indeed needed for carboxydofrophy, but high expression is detrimental for

growth on CO as it results in overreduction of the Fd pool. CODH downregulation therefore complements Ech2 upregulation in CO-1.

21. Line 318: Monofunctional CODHs are not using CO directly?

They are, the sentence was slightly modified (line 365).

22. Line 367 – “A single isolate...” – Why was only a single isolate sequenced?

The first one that grew robustly on CO was selected for further characterization.

23. Lines 385 – 387 – The hypothesis about potential changes in Ech2 subunit expression levels is understandable, but seems out of place amidst a section on chromosomal rearrangements.

Not if Ech2 subunit genes expression depends on chromosomal rearrangements. By placing *ech2* genes close to the chromosomal replication terminus, the large inversion might have resulted in a decrease of expression of *ech2* subunits compared to the initial KI locus. The promoter driving the new *ech2* genes is an endogenous promoter that, in our datasets, drives the expression of its cognate native gene (S-layer protein) to very high levels (about 45,000 TPM). On the other hand, *ech2* subunits genes in Ech2KI display a level of expression approximately 10 times lower, despite using the same promoter. It is therefore possible that the inversion happened to reduce the level of expression of *ech2* genes to a level optimizing the microbe's fitness on CO (that is, the selective pressure).

This was discussed in the text (lines 473-485).

24. Lines 390 – 391 – “...could modulate carbon metabolism.” Could as in have been shown to regulate carbon metabolism? If yes, in which organisms?

The two references reported here directly mention *E. coli* and *B. subtilis*.

25. Line 398 – “...multiple loci...” Please state how many.

The number (6) was added in the text (line 455). This is described in Supplementary Table S4 that is referenced here.

26. Figure 7c and 7d – Please discuss the lower max OD of Ech2KI compared to CO-1, as this is important. Especially in light of multitude of genomic rearrangements in the KI strain. Also, the figure could benefit from making the bar charts larger for easier inspection. Please specify in the figure legend that the red-colored part of the chromosome on 7d marks the inversion event.

The main point of figure 7c was to show how the Ech2KI strain was generated and that it can grow under CO as sole carbon/energy source (as now stated lines 416-419). The difference in the OD measured after 3 days could be linked to a difference in growth rate.

The bar plots and the legend were modified.

27. Lines 425 – 426 – How would you explain such a drastic difference in Td between the CO-1 strain and the previously CO-adapted *T. kivui* by Weghoff & Müller? Please address in the text.

In Weghoff and Müller, the strain was adapted to grow mixotrophically on CO + YE, whereas we adapted the strain to grow on CO only. As a result, our selective conditions were harsher, and the resulting strain possibly more robust. A statement was added lines 491-494.

28. Lines 454 – 455 – “...our study demonstrated Ech2 KI could kickstart growth on CO...” A little difficult to claim that, as the KI strain carries so many additional mutations. Please dial down, suggestion to “...demonstrated that Ech2 KI could lead to growth on CO” .

Because selection for carboxydrotrophic growth yielded a strain with a correct integration of the *ech2* overexpression cassette (as opposed to selection for uracil prototrophy, Figure 7), we do not believe this statement needs to be dialed down, particularly in the light of the novel results reported in this revised manuscript.

29. Line 492-495: Would be relevant to add DOI: 10.1016/j.cej.2022.137678 as reference and expand sentence

The paper was added, as well as a hypothesis as to how their results could be compatible to ours (lines 581-590)

30. Lines 528-545: It can be assumed that all this text applies to *T. kivui*, so best to specify that to avoid the confusion whether they also apply for *E. coli*

The document was emended (lines 621-633).

31. Line 544: Define shaking speed

Shaking is defined in the fermentation in bioreactor subsection (lines 662-666).

32. Line 556: Specify/define “high growth”

A statement was added in the text (line 658).

33. Line 557: Specify if colonies were grown autotrophically

They were not and this was added lines 658-659.

34. Line 579: State the dilution rates used

A statement was added in the text (lines 681-682).

35. Lines 583-587: Specify mobile phase, flow rate, column temperature etc.

A statement was added in the text (lines 687-688).

6. Line 589: Define how steady-state was determined

Constant value for OD, acetate titer, and gas uptake/production rates for 4-5 volume changes (lines 693-694).

37. Line 603-...: Specify parameters for pelleting (how much biomass used, centrifugation etc)

A statement was added lines 708-709.

38. Line 611: Specify read length

A statement was added line 717. Further details in Supplementary Table S6.

39. Line 633: “was” missing between “sequence” and “manually”

The sentence was removed.

40. Line 656: “was” missing after “chromosome”; “these” instead of “this”

“Was” was added, “this” was changed to “these” (line 756).

41. Line 686 - Genotyping – Please add more details to the semi-quantitative PCR approach. How much DNA was used, number of cycles, etc. or reference the protocol used in this work.

This information was added lines 793-794.

42. Line 694: “pelleted” instead of “pellets”; specify parameters for pelleting (how much biomass used, centrifugation etc)

A statement was added line 797.

43. Line 697: Specify conditions for lysis

A statement was added lines 800-801.

44. Line 698: Please note how was RNA integrity evaluated?

Via a fragment analyzer. A statement was added line 803.

45. Line 764: Accession number missing

The accession number were added in the text (lines 867-871).

Reviewer #5 (Remarks to the Author):

Response to reviewers

We would like to express our sincere thanks to all reviewers for their positive evaluation of our revised manuscript. A point-by-point response is hereafter provided for Reviewer #4.

REVIEWER COMMENTS

Reviewer #1 (Remarks to the Author):

The authors have adequately addressed this reviewer's questions and provided the necessary revisions to strengthen the manuscript. I thus endorse the manuscript for publication in Nature Communications.

Reviewer #2 (Remarks to the Author):

I have reviewed the resubmission of A megatransposon drives the adaptation of *Thermoanaerobacter kivui* to carbon monoxide. The authors have addressed my concerns as well as those of the other reviewers. I recommend that the manuscript be published.

Reviewer #3 (Remarks to the Author):

The authors have fully addressed the suggested comments by incorporating several detailed descriptions and supplements: discussing CO metabolism of *T. kivui* in the introduction, its characteristics as a biocatalyst, and providing a detailed explanation of plausible mechanisms for transposition in the discussion. Additionally, they have moderately described TnCO-1 maintenance, clarified the rationale for the TnCO-1 elimination method, and conducted additional transcriptomics experiments and analyses that support the carboxydrotrophic phenotype of *T. kivui*. The revised manuscript has improved in clarity and is now better supported by additional data and explanations.

Reviewer #4 (Remarks to the Author):

The authors have addressed all but one comment adequately. In addition, please check the reference to Supplementary Figure S3. After addressing the latter, I support publication of this work. I congratulate the authors on this landmark work!

General comments

1. The results and claims made about the effect of Ech2 knock-in are not convincing currently. Firstly, no experimental data currently show elevated expression of Ech2 in the KI strain. Please provide qPCR or protein expression results to show elevated Ech2 expression in the Ech2KI strain vs the PyrEKIpop culture (WT is not a correct control as Ech2KI is supplied with uracil that can affect growth and Ech2KI went through sub-culturing). It would also be interesting to see Ech2 expression in Ech2KI vs CO-1. Secondly, Ech2KI mutation results in Lines 383-420 are presented vs the WT (non-uracil medium). Growth uracil and sub-culturing of the population and selection during one month prior to clone isolation could introduce mutations not triggered by Ech2 knock-in (as was the case when H-2, H-1 and CO-1 were isolated). Thus to support the claims about potential roles of the mutations, the authors would need to compare Ech2KI mutation data vs PyrEKIpop or isolates that have been sub-cultured/selected on uracil for the same period.

Please note that uracil was not used during the selection procedure, as we conversely selected for uracil prototrophy (this confusion likely stemmed from the “minus” sign used in the figure, that could be interpreted as a “dash” – the figure was edited to avoid this confusion). The PyrEKIpop culture “died” during co-selection on CO + CDM. However, we have an isolated clone PyrEKI from the direct plating selection step. We cultivated that clone, Ech2KI and CO-1 using syngas in serum bottles until the log phase, and performed RNA-seq. The resulting data showed unambiguously that the *ech2*-related genes were upregulated in Ech2KI (Supplementary Figure S4, Supplementary Table S7) and, even more so, that *ech2* overexpression is the main difference arising between Ech2KI and PyrEKI at the transcriptional level. We added these results at lines 473-485.

It seems line 476 mistakenly refers to Supplementary Figure S3, instead of Supplementary Figure S4.

Figure S4 should indeed be referenced here. We have emended the text accordingly (line 476).

28. Lines 454 – 455 – “...our study demonstrated Ech2 KI could kickstart growth on CO...” A little difficult to claim that, as the KI strain carries so many additional mutations. Please dial down, suggestion to “...demonstrated that Ech2 KI could lead to growth on CO” . Because selection for carboxydrotrophic growth yielded a strain with a correct integration of the *ech2* overexpression cassette (as opposed to selection for uracil prototrophy, Figure 7), we do not believe this statement needs to be dialed down, particularly in the light of the novel results reported in this revised manuscript.

The novel results do not change the fact that the KI strain carries many additional mutations. Also in the new Supplementary Figure S4, there are genes that do not belong to the *ech2* locus but expressed equally highly – orange dots among the green *ech2* locus “cloud”. So it would be fair to dial down.

The reviewer has a point that indeed some mutations arose in the Ech2KI strain. In Figure 7c, we show that cells transformed with an *ech2* overexpression cassette were able to grow on CO (both replicates grew). This was not the case for the negative control (both replicates did not grow), despite the fact that they were given the same opportunity to accumulate random mutations. The main difference between these conditions is the integration of the cassette, and *ech2* overexpression was indeed checked during our first revision of this manuscript. Therefore, it is safe to say the overexpression of *ech2* subunit genes could kick-start growth on CO, that is, that the overexpression of *ech2* gave the initial impulse to allow cells to grow on CO. Other mutations in the *ech2* overexpression strain might have further enhanced growth on CO later (e.g., by refining *ech2* expression level, as discussed in the text) but the necessary impulse originated from the overexpression.

In addition, please identify those most up-regulated (orange) genes part of the *ech2* cloud on the volcano plot as well as the 5 most down-regulated genes.

The labels were added. Most of these correspond to genes with unknown functions (in which case only the locus tag is reported).

Reviewer #5 (Remarks to the Author):

I co-reviewed this manuscript with one of the reviewers who provided the listed reports. This is part

of the Nature Communications initiative to facilitate training in peer review and to provide appropriate recognition for Early Career Researchers who co-review manuscripts.